# Large Language Models and Multimodal Retrieval for Visual Word Sense Disambiguation

**Anastasia Kritharoula**
anaskrith@gmail.com

**Maria Lymperaiou**
marialymp@islab.ntua.gr

**Giorgos Stamou**
gstam@cs.ntua.gr

Artificial Intelligence and Learning Systems Laboratory
School of Electrical and Computer Engineering
National Technical University of Athens

## Abstract

Visual Word Sense Disambiguation (VWSD) is a novel challenging task with the goal of retrieving an image among a set of candidates, which better represents the meaning of an ambiguous word within a given context. In this paper, we make a substantial step towards unveiling this interesting task by applying a varying set of approaches. Since VWSD is primarily a text-image retrieval task, we explore the latest transformer-based methods for multimodal retrieval. Additionally, we utilize Large Language Models (LLMs) as knowledge bases to enhance the given phrases and resolve ambiguity related to the target word. We also study VWSD as a unimodal problem by converting to text-to-text and image-to-image retrieval, as well as question-answering (QA), to fully explore the capabilities of relevant models. To tap into the implicit knowledge of LLMs, we experiment with Chain-of-Thought (CoT) prompting to guide explainable answer generation. On top of all, we train a learn to rank (LTR) model in order to combine our different modules, achieving competitive ranking results. Extensive experiments on VWSD demonstrate valuable insights to effectively drive future directions.

## 1 Introduction

Visual word sense disambiguation (VWSD) is a recently introduced challenging task where an ambiguous target word within a given context has to retrieve the proper image among competitive candidates (Raganato et al., 2023). For example, the phrase *andromeda tree* contains the ambiguous target word *andromeda* accompanied by the context *tree* which resolves this ambiguity. Out of the 10 candidates presented in Fig. 1, a VWSD framework attempts to retrieve the ground truth image, denoted with colored border.

Even though VWSD is essentially a text-image retrieval task, there are some fundamental differences. First of all, the context given for an ambiguous word is minimal, most often limited to a single

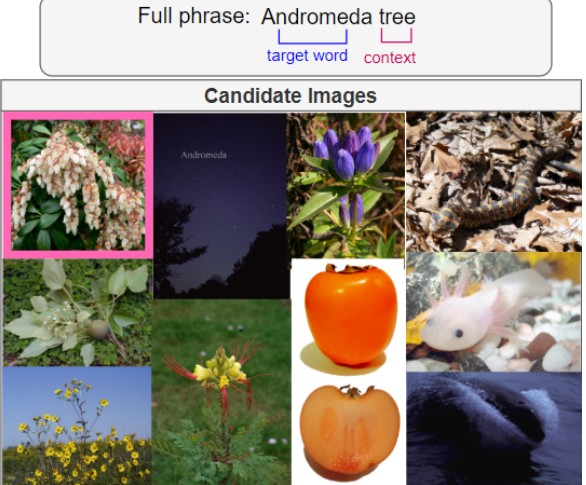

Figure 1: An example of the VWSD task.

word, upon which a retrieval module should rely to retrieve the proper candidate. Additionally, the candidate images themselves pose significant challenges; for example, by observing the candidates of Fig. 1, which are related to several meanings of the ambiguous word *andromeda* (can be either a constellation, fish species, tree, reptile etc), a successful retrieval system should be fine-grained enough and highly sensitive to the contextualization of the ambiguous target word. In that case, the context word *tree* should be prominent enough -from the perspective of the retrieval module- to resolve ambiguity. At the same time, this retrieval module should not excessively rely on the *tree* context: images containing flowers and green grass induce some expected visual bias in the retrieval process, thus the image having the highest probability of containing a *tree* may be selected, ignoring the ambiguous *andromeda* attribute. Of course, there is also the possible scenario that a retrieval model has never been trained on the ambiguous word at all: the rarity of the concepts present in the target word vocabulary increases the probability of exclusively relying on the well-known context

word, resulting in significant randomness during selection. To this end, VWSD trustworthiness also arises as a critical point, raising the need for explainable solutions.

In this work, we showcase a vast variety of implementations for VWSD. Multiple experiments are conducted for each of the implemented approaches, achieving one of the first extensive contributions to this interesting task:

- We exploit Large Language Models (LLMs) as knowledge bases to enrich given full phrases, so that the target word is disambiguated by incorporating more context, addressing even cases that the ambiguous word is unknown to the retrieval module.

- We convert VWSD to a unimodal problem: retrieval (text-to-text and image-to-image) and question-answering (QA) to fully explore the capabilities related models have to offer.

- Features extracted from the aforementioned techniques are used to train a learning to rank model, achieving competitive retrieval results.

- Chain-of-Thought (CoT) prompting is leveraged to guide answer generation, while revealing intermediate reasoning steps that act as explanations for retrieval.

Our code can be found at https://github.com/anastasiakrith/multimodal-retrieval-for-vwsd/.

## 2 Related work

**Text-image retrieval** has been revolutionized since adopting the popular Transformer framework (Vaswani et al., 2017) to further incorporate the visual modality. This transition towards multimodality was addressed by incorporating one additional transformer stream to process images (Tan and Bansal, 2019; Lu et al., 2019), extending the BERT (Devlin et al., 2019) architecture. Most recent approaches (Kim et al., 2021; Huang et al., 2021; Wang et al., 2021) improved upon those primordial works by utilizing a single encoder for both images and language, therefore minimizing the number of trainable parameters, and consequently improving the performance of several VL tasks, including multimodal retrieval. A significant milestone was the adoption of contrastive learning for text-image representations, a technique that is followed by CLIP (Radford et al., 2021) and ALIGN (Jia et al., 2021).

Visual Word Sense Disambiguation (VWSD) (Raganato et al., 2023) is only recently introduced as part of the SemEval 2023 challenge. So far, the concurrent work of Dadas (2023) can act as a measure of comparison to ours. We adopt and extend some ideas presented in their paper, while further expanding the experimental suite to cover variable approaches to the task. Moreover, we step upon the usage of LLMs for VWSD as in Kritharoula et al. (2023) to address both performance improvement and explainability aspects.

**LLMs as knowledge bases** is a core idea followed throughout our paper, as enriching the short phrases of the VWSD dataset can facilitate target word disambiguation, and thus improve retrieval. Traditionally, knowledge enhancement for disambiguation was performed via knowledge graphs (Feng et al., 2020; Nedelchev et al., 2020). The usage of knowledge graphs was also favored for knowledge enhancement of multimodal tasks (Lymperaiou and Stamou, 2022). Nevertheless, Large Language Models (LLMs) as knowledge bases (LLM-as-KB) (Petroni et al., 2019; AlKhamissi et al., 2022) is a novel paradigm, presenting some interesting capabilities compared to traditional knowledge graphs. Knowledge retrieval from LLMs is implemented via prompting (Liu et al., 2021), which attempts to appropriately trigger the LLM in order to provide the fact requested. To this end, recent VL breakthroughs favor the LLM-as-KB paradigm for knowledge enhancement, even though some unresolved shortcomings may be inherited (Lymperaiou and Stamou, 2023). We opt for incorporating the LLM-as-KB paradigm within our experimentation to investigate performance gains over knowledge-free baselines, following Kritharoula et al. (2023).

## 3 Method

We followed 6 approaches to investigate the VWSD task from several different perspectives. All our approaches were tested exclusively on English.

**1. Image-Text similarity baseline** We implement a simple multimodal (VL) retrieval baseline to evaluate the capabilities of existing pre-trained VL transformers on the VWSD task. VL transformers provide joint embedding representations for text phrases $t$ and candidate images $i$, and the image representation achieving highest cosine similarity score $score(t, i) = max(sim(t, i))$ with respect to

the text embedding is selected. We utilize CLIP with ViT (Dosovitskiy et al., 2021) base encoder, as well as with ViT large encoder (denoted as CLIP-L). ALIGN (Jia et al., 2021) is also used for text-image retrieval. We also leverage several versions of BLIP (Li et al., 2022), namely $BLIP_C$ and $BLIP$-$L_C$ (pre-trained on COCO (Lin et al., 2015) and using ViT base/ViT large as backbone encoders respectively), as well as $BLIP_F$ and $BLIP$-$L_F$ (pre-trained on Flickr30k (Young et al., 2014)). More details are provided in Appendix G. We also experiment with incorporating the penalty factor $p(i)$ described in Dadas (2023) to modulate the retrieval preference of images that present high similarity scores $sim(t, i)$ to multiple phrases $t$. In this case, the similarity score obeys to the following:

$$score(t, i) = sim(t, i) - p(i) \qquad (1)$$

**2. LLMs for phrase enhancement** We employ a variety of LLMs as knowledge bases to enhance the short phrases $t$ with more detail in a zero-shot fashion (Kritharoula et al., 2023) and thus facilitate VL retrieval described in the previous paragraph. All prompts to LLMs presented in Tab. 1 are designed upon manually crafted templates, based on the intuition that instructively requesting specific information from the model has been proven to be beneficial (Kojima et al., 2023). We select LLMs up to the scale that our hardware allows, or otherwise accessible via public APIs; specifically, GPT2-XL (1.5B parameters) (Radford et al., 2019), BLOOMZ-1.7B & 3B (Muennighoff et al., 2023), OPT-2.7B & 6.7B (Zhang et al., 2022), Galactica 6.7B (Taylor et al., 2022), and the 175B parameter GPT-3 (Brown et al., 2020) and GPT-3.5-turbo[1] comprise our experimental set for phrase enhancement. The difference in the number of parameters is viewed as significant in order to indicate whether scale matters to elicit disambiguation capabilities of LLMs. We denote as $t_e$ the LLM-enhanced phrases. Similar to the baseline case, a penalty factor $p(i)$ can be included, adjusting the LLM-enhanced retrieval score as:

$$score(t_e, i) = sim(t_e, i) - p(i) \qquad (2)$$

**3. Image captioning for text retrieval** We leverage the merits of unimodal retrieval by exploiting state-of-the-art image captioning transformers to

---

[1]https://platform.openai.com/docs/models/gpt-3-5

| Prompt name | Prompt template |
|---|---|
| exact | "<phrase>" |
| what_is | "What is <phrase>?" |
| describe | "Describe <phrase>." |
| meaning | "What is the meaning of <phrase>?" |

Table 1: Prompts for phrase enhancement via LLMs.

convert images $i$ to textual captions $c_i$. Specifically, the captioning models used are BLIP Captions (Li et al., 2022) with ViT-base encoder (BLIP-L Captions denotes building upon ViT-large), as well as GiT (Wang et al., 2022) (with ViT-base) and GiT-L (with ViT-large). For all BLIP and GiT variants we attempt both beam-search multinomial sampling with 5 beams to obtain $k$=10 captions per image $i$, as well as greedy search. We symbolize as $c_i^k$ the $k$-th caption for an image $i$, as obtained from beam search (greedy search returns a single caption). In the case of beam search, the 10 captions are post-processed, as some of them are identical or substrings of longer ones.

We explore two options in obtaining embedding representations for the captions $c_i$ and the phrases $t$. In the first case, embedding representations are obtained using the same VL transformers as in multimodal retrieval. In the second case, we utilize a variety of purely textual sentence transformers that are fine-tuned for semantic similarity (Reimers and Gurevych, 2019). Then, for both cases, we use cosine similarity or euclidean/manhattan [2] distance to calculate the $score(t, c_i^k)$, thus retrieving the most similar caption embedding to each phrase embedding. Experiments with and without LLM-based phrase enrichment were conducted.

**4. Wikipedia & Wikidata image retrieval** Image-to-image retrieval is another way to approach the VWSD task via unimodal representations. For this reason, following the idea of (Dadas, 2023) we exploit the Wikipedia API to retrieve all relevant articles corresponding to the given phrase $t$, and we keep the primary image $i_w$ from each article. Consequently, we post-process the retrieved image set by considering a maximum of $k$=10 Wikipedia images per $t$. The same process is repeated for Wikidata (Vrandečić and Krötzsch, 2014). We obtain embedding representations for the retrieved images $i_w$, as well as for the candidate images $i$, using the same VL transformers as in mul-

---

[2]Distance metrics are also denoted as 'similarity metrics' throughout this paper

timodal retrieval. Finally, we search for the embeddings lying closer together in the embedding space (using cosine similarity or euclidean/manhattan distance) according to $score(i_w, i)$.

**5. Learn to Rank** Similarly to Dadas (2023), we implement a lightweight Learning to Rank (LTR) model that harnesses features extracted from our aforementioned experiments. LGBMRanker with lambdarank objective[3], implemented upon the LightGBM gradient boosting framework (Ke et al., 2017), is selected as the LTR module.

The selected input features for the LTR model represent relationships between each given phrase and the candidate images extracted from the previous 4 approaches. Specifically, the following steps (a)-(e) are selected to craft features corresponding to the baseline case. In a similar way, the steps a-e are repeated for $score(t_e, i)$ (LLM enhancement), $score(t, c_i^k)/score(t_e, c_i^k)$ (caption-phrase retrieval/enhanced caption-phrase retrieval) and $score(i_w, i)$ (image retrieval). We train the LTR module on several combinations of the designed features; also, different similarity (cosine) and distance (euclidean/manhattan) scores are attempted within these combinations, while the contribution of considering $p(i)$ is evaluated, both in the baseline VL retrieval (eq. 1), as well as in the LLM-enhanced VL retrieval module (eq. 2).

---

(a) $score(t, i)$

(b) $max(score(t, i))$

(c) $mean(score(t, i))$

(d) $difference\ a\text{-}b$

(e) $difference\ a\text{-}c$

---

In order to further advance LTR performance, we attempt to combine features from enriched phrases $t_e$ derived from different LLMs.

**6. Question-answering for VWSD and CoT prompting** Based on the zero-shot question-answering setting of Kritharoula et al. (2023), we incorporate the given phrase $t$ within a question template, providing 10 answer choices from A to J that correspond to extracted captions $c_i$ from the candidate images $i = A, B, ..., J$ for each $t$. In the case that beam search was used during captioning, all $k$=10 captions for each $i$ are concatenated and separated by comma to form each answer choice.

[3]LGBMRanker docs

Moreover, based on Kojima et al. (2023) we utilize chain-of-thought (CoT) prompts to obtain explanations regarding the reasoning process the LLM follows when selecting an answer. All resulting QA prompts (Tab. 2) are provided to GPT-3.5-turbo.

## 4 Experimental results

The VWSD dataset is comprised of 12869 training samples and 463 test samples; each sample contains 10 images (Appendix A). All our approaches are evaluated on the VWSD test split using Accuracy and Mean Reciprocal Rank (MRR) metrics.

**LLMs for phrase enhancement** In Tab. 3, we present results regarding LLM-based phrase enhancement involving all VL retrieval models (with and without penalty $p(i)$). Baselines refer to VL retrieval with non-enhanced phrases $t$. Additional quantitative results including more LLMs, as well as qualitative examples of how knowledge enhancement benefits disambiguation and thus retrieval are provided in Appendix C. We can easily observe that different prompts contribute towards better results per LLM, while BLIP-L$_F$ is the most successful VL retriever; however, ALIGN (with penalty) achieves top performance, together with GPT-3 and "meaning" as the prompt. In general, large scale LLMs (GPT-3, GPT-3.5-turbo) are able to clearly surpass the respective non-enhanced baselines in most cases.

Interestingly, for smaller LLMs (up to 6.7B parameters), scaling-up does not necessarily imply advanced performance: OPT-6.7B (Tab. 12) and Galactica-6.7B (Tab. 13) fall below their smaller BLOOMZ-3B competitor when the same prompts are used. Nevertheless, contrary to GPT-3 and GPT-3.5-turbo, the phrase enhancement that few-billion parameter LLMs can offer in total is only marginal in the best case, and sometimes even fail to compete against their non-enhanced baselines, indicating that they do not contain the necessary knowledge to enrich rare target words with respect to their given context. Therefore, our LLM-enhancement analysis reveals that the necessary enrichment for VWSD may be only achieved when employing large-scale LLMs, most probably being on par with other emergent LLM abilities (Wei et al., 2023).

**Image captioning** In Tab. 4 we present results on text retrieval between extracted image captions $c_i$ and given phrases $t$, which are achieved us-

| Prompt name | QA Prompt template |
|---|---|
| think (greedy) | "Q: What is the most appropriate caption for the <context>? Answer choices: (A) <caption for image 1> (B) <caption for image 2> ... A: Let's think step by step. " |
| think (beam) | "Q: What is the most appropriate group of captions for the <context>? Answer choices: (A) <captions for image 1 (separated with comma)> (B) <captions for image 2> ... A: Let's think step by step. " |
| CoT | "<think_prompt> <response of llm with think prompt> Therefore, among A through J, the answer is " |
| no_CoT (greedy) | "Q: What is the most appropriate caption for the <context>? Answer choices: (A) <caption for image 1> (B) <caption for image 2> ... A: " |
| no_CoT (beam) | "Q: What is the most appropriate group of captions for the <context>? Answer choices: (A) <captions for image 1> (B) <captions for image 2> ... A: " |

Table 2: QA prompts for gpt-3.5-turbo containing given phrase and extracted image captions.

ing Manhattan distance as a similarity measure. ALIGN and XLM distilroberta base (Reimers and Gurevych, 2019) perform the caption-phrase retrieval, representing textual representations via VL models and purely linguistic semantic similarity models respectively. More text-to-text retrieval results in Appendix D. The No-LLM row of Tab. 4 refers to the case that no phrase enhancement is performed, while the rest of the cases correspond to prompts designed as per Tab. 1 towards enhanced phrases $t_e$. In all results presented, GPT-3 is selected as the LLM to be prompted, as it demonstrated superior knowledge-enhancement performance. We observe that LLM-based enhancement offers performance boosts to text-to-text retrieval compared to the No-LLM baseline in most cases; nevertheless, it still stays behind the best performance so far (acc 72.57%, MRR 82.29%) achieved by GPT-3-enhanced VL retrieval. We assume this is because of the information loss induced when converting from the visual to the textual modality during captioning.

Another observation is that VL transformers perform better in producing *textual* embeddings compared to sentence similarity embeddings, even though the latter have been explicitly fine-tuned on semantic textual similarity. This observation is cross-validated by examining more purely textual semantic similarity transformers in Appendix D.

**Wikipedia & Wikidata image retrieval** In Tab. 5 results regarding image-to-image retrieval between candidates $i$ and web retrieved images $i_w$ are presented. Out of the 463 samples of the test set, Wikipedia API and Wikidata API returned results for 460 and 324 phrases respectively. Even best results for image-to-image retrieval are not competent against our previous approaches; we assume that exclusively visual representations are not expressive enough to distinguish fine-grained

details between semantically related candidates.

**Learn to rank** In Tab. 6 we exhibit results using ALIGN as the VL retriever. The presented feature combinations involve the following: (1) Baseline features: the choice for incorporation (or not) of penalty $p(i)$ in $score(t, i)$ for the VL retrieval; (2) LLM-enhancement features: the prompt to produce enhanced phrases $t_e$ (or an ensemble of prompts leading to multiple $t_e$) and the choice for incorporation (or not) of $p(i)$ in $score(t_e, i)$; (3) Text retrieval features: the captioner to generate $c_i$, together with the text embedding model and the similarity measure (cosine/euclidean/manhattan) for text-to-text retrieval, as well as the phrase (original $t$, or enhanced $t_e$, or an ensemble of enhanced phrases $t_e$ derived from different LLMs); (4) Image retrieval features: image embedding model and similarity measure (cosine/euclidean/manhattan) for image-to-image retrieval. Additional results for the LTR are provided in Appendix E.

For all the experiments of Tab. 6 we used the following hyperparameter configuration: n_estimators: 500, early_stopping: 100, learning rate: 0.03, feature_fraction: 0.25, max_bin: 100, min_child_samples: 50 and reg_alpha: 0.05. An 80-20 train/validation split was followed, allocating 2514 samples in the validation set.

The ablations of feature combinations presented in Tab. 6 are highly informative, indicating that different features pose a varying influence on the final result, while the amount of incorporated features is also significant. In general, incorporating LLM-based phrase enhancement in LTR is highly beneficial, offering optimal metric results compared to other feature combinations, or our other approaches presented in Tab. 3, 4. Overall best results are achieved when including all features (colored instances of Tab. 6). This is an interesting observation since standalone text retrieval (Tab. 4)

| | | CLIP | | CLIP-L | | ALIGN | | BLIP$_C$ | | BLIP-L$_C$ | | BLIP$_F$ | | BLIP-L$_F$ | |
|---|---|---|---|---|---|---|---|---|---|---|---|---|---|---|---|
| | | acc. | MRR | acc. | MRR | acc. | MRR | acc. | MRR | acc. | MRR | acc. | MRR | acc. | MRR |
| **With penalty** | | | | | | | | | | | | | | | |
| **Baseline** | | 63.28 | 76.27 | 62.85 | 76.24 | 68.90 | 80.00 | 60.90 | 74.33 | 64.58 | 77.51 | 60.47 | 73.87 | 69.76 | 80.42 |
| OPT-2.7B | exact | 62.85 | 76.00 | 62.85 | 75.93 | 68.68 | 79.89 | 61.12 | 74.46 | 64.58 | 77.41 | 60.26 | 73.73 | **69.76** | **80.36** |
| | what_is | 60.98 | 74.85 | 66.30 | 78.10 | 63.28 | 75.95 | 60.91 | 74.43 | 66.31 | 77.86 | 57.24 | 71.15 | 67.60 | 78.58 |
| | describe | 61.05 | 74.75 | 66.08 | 78.14 | 64.79 | 77.62 | 61.77 | 74.73 | 66.31 | 77.57 | 57.67 | 71.48 | 68.03 | 79.03 |
| | meaning | 62.15 | 75.60 | 65.25 | 77.45 | 65.66 | 77.54 | 61.99 | 75.35 | 63.93 | 76.88 | 58.32 | 71.65 | 65.44 | 77.69 |
| BLOOMZ-3B | exact | 61.26 | 74.59 | 62.99 | 76.18 | 66.52 | 78.36 | 60.48 | 73.13 | 63.28 | 76.00 | 57.02 | 71.23 | 65.66 | 77.49 |
| | what_is | 64.36 | 76.82 | 68.25 | 79.82 | 67.39 | 78.72 | 61.34 | 74.94 | 66.95 | 78.47 | 59.61 | 73.35 | **68.47** | **79.58** |
| | describe | 62.01 | 75.38 | 65.28 | 78.07 | 66.09 | 78.60 | 62.85 | 75.65 | 67.39 | 78.71 | 57.24 | 71.72 | 67.82 | 79.20 |
| | meaning | 65.58 | 77.96 | 67.32 | 78.76 | 68.47 | 79.14 | 63.71 | 76.52 | 66.31 | 78.55 | 59.40 | 73.60 | 68.03 | 79.26 |
| GPT-3.5 | exact | 58.86 | 72.09 | 60.18 | 72.73 | 62.42 | 74.43 | 57.02 | 70.78 | 59.18 | 72.32 | 52.92 | 67.40 | 63.07 | 74.65 |
| | what_is | 66.52 | 78.81 | 69.35 | 80.51 | 70.41 | 81.42 | 67.60 | 78.56 | 68.47 | 79.67 | 60.91 | 74.30 | **71.71** | **82.02** |
| | describe | 67.32 | 78.95 | 69.28 | 80.31 | 73.22 | 82.73 | 69.33 | 79.90 | 70.41 | 80.80 | 59.83 | 73.65 | 70.63 | 81.29 |
| | meaning | 67.76 | 79.76 | 69.06 | 80.55 | 70.41 | 81.38 | 66.52 | 78.59 | 66.52 | 79.16 | 58.53 | 73.31 | 69.98 | 81.46 |
| GPT-3 | exact | 61.98 | 74.90 | 64.07 | 76.58 | 66.52 | 78.37 | 60.48 | 73.99 | 64.15 | 76.58 | 59.61 | 72.91 | 65.23 | 77.06 |
| | what_is | 67.92 | 79.27 | 70.73 | 81.57 | 71.71 | 82.27 | 68.25 | 78.93 | 68.90 | 79.91 | 60.48 | 74.24 | 69.11 | 80.25 |
| | describe | 68.25 | 79.40 | 68.72 | 80.26 | 72.57 | 82.52 | 64.58 | 76.75 | 68.25 | 79.35 | 61.34 | 74.03 | 69.33 | 80.47 |
| | meaning | 68.07 | 80.08 | 69.84 | 81.56 | **74.95** | **84.09** | 66.74 | 78.37 | 71.71 | 81.55 | 62.63 | 75.55 | **72.35** | **82.28** |
| **Without penalty** | | | | | | | | | | | | | | | |
| **Baseline** | | 59.18 | 72.94 | 60.69 | 74.42 | 65.66 | 77.48 | 57.24 | 72.07 | 61.34 | 75.88 | 57.67 | 71.96 | 65.01 | 77.86 |
| OPT-2.7B | exact | 58.96 | 72.77 | 60.26 | 74.15 | 65.66 | 77.48 | 57.45 | 72.19 | 61.12 | 75.77 | 57.24 | 71.68 | **65.01** | **77.90** |
| | what_is | 58.31 | 72.91 | 62.75 | 75.47 | 61.12 | 73.94 | 59.83 | 73.13 | 61.12 | 74.54 | 53.35 | 68.71 | 63.50 | 76.22 |
| | describe | 59.08 | 72.95 | 63.89 | 76.31 | 62.20 | 75.80 | 59.83 | 73.28 | 62.20 | 75.17 | 54.43 | 69.86 | 63.28 | 76.28 |
| | meaning | 58.19 | 72.97 | 62.99 | 75.79 | 64.58 | 76.48 | 59.18 | 73.38 | 60.26 | 74.70 | 54.86 | 69.43 | 62.42 | 75.86 |
| BLOOMZ-3B | exact | 56.93 | 71.53 | 59.52 | 73.78 | 63.93 | 76.15 | 58.10 | 71.77 | 59.61 | 74.06 | 54.86 | 69.66 | 61.12 | 74.99 |
| | what_is | 62.20 | 75.39 | 65.66 | 77.88 | 62.85 | 75.51 | 61.34 | 74.35 | 65.01 | 77.32 | 57.24 | 71.85 | **68.03** | **79.12** |
| | describe | 60.04 | 73.83 | 62.88 | 76.11 | 63.50 | 76.35 | 60.48 | 73.87 | 62.85 | 76.06 | 54.86 | 70.48 | 65.66 | 77.64 |
| | meaning | 61.69 | 75.51 | 64.94 | 77.17 | 66.31 | 77.62 | 61.77 | 74.92 | 62.42 | 76.27 | 57.02 | 71.79 | 65.23 | 77.21 |
| GPT-3.5 | exact | 56.89 | 69.85 | 57.11 | 70.36 | 60.48 | 72.15 | 54.43 | 68.33 | 56.80 | 70.42 | 51.40 | 65.68 | 58.32 | 71.11 |
| | what_is | 65.00 | 77.11 | 65.87 | 78.11 | 67.82 | 79.52 | 64.15 | 75.91 | 65.87 | 77.78 | 58.10 | 72.32 | 68.03 | 79.36 |
| | describe | 65.80 | 77.26 | 66.67 | 78.42 | 70.84 | 81.16 | 65.44 | 77.57 | 69.11 | 80.20 | 58.96 | 72.66 | 67.60 | 79.47 |
| | meaning | 65.14 | 77.61 | 67.10 | 79.07 | 68.47 | 79.87 | 63.93 | 77.05 | 65.66 | 78.33 | 63.93 | 72.23 | **68.25** | **80.17** |
| GPT-3 | exact | 59.88 | 73.38 | 61.68 | 74.91 | 64.79 | 76.27 | 58.96 | 71.92 | 60.48 | 74.02 | 55.72 | 70.34 | 62.42 | 75.04 |
| | what_is | 66.51 | 77.62 | 68.15 | 79.38 | 69.55 | 80.22 | 63.28 | 75.56 | 65.01 | 77.40 | 56.59 | 71.54 | 67.82 | 79.03 |
| | describe | 67.30 | 78.50 | 68.25 | 79.81 | 71.27 | 81.21 | 63.93 | 75.81 | 66.31 | 77.74 | 58.96 | 72.62 | 67.17 | 78.93 |
| | meaning | 66.52 | 78.32 | 68.96 | 80.26 | 72.57 | 82.29 | 65.87 | 77.56 | 69.55 | 80.26 | 60.26 | 74.26 | 70.41 | 81.09 |

Table 3: Results for zero-shot LLM-based enhancement. **Colored** instances denote overall best results per metric, while **bold** numbers indicate best results for each LLM.

and image retrieval (Tab. 5) experiments did not provide competitive metric results; nevertheless, considering the respective features in LTR training benefits performance. Moreover, ensembling of features is highly beneficial. This applies on both ensembling the LLM-enhanced prompt features (e.g. *all prompts* combines features from $t_{exact}$, $t_{what\_is}$, $t_{describe}$, $t_{meaning}$), as well as ensembling phrase features for text-to-text retrieval (*all* $t_e$ +

$t$ refers to combining features from all the aforementioned 4 enhancements, plus the original given phrase $t$). As demonstrated in Tab. 6, most ensemble feature combinations help surpassing baselines and other implementations (Dadas, 2023).

The implemented LTR module is computationally efficient, as it only requires a CPU for training, while achieving state-of-the-art performance (Tab. 9). The clear pattern that arises from the ablation

| | Greedy | | | | | | | | Beam | | | | | | | |
|---|---|---|---|---|---|---|---|---|---|---|---|---|---|---|---|---|
| | **BLIP** | | **BLIP-L** | | **GiT** | | **GiT-L** | | **BLIP** | | **BLIP-L** | | **GiT** | | **GiT-L** | |
| | acc. | MRR | acc. | MRR | acc. | MRR | acc. | MRR | acc. | MRR | acc. | MRR | acc. | MRR | acc. | MRR |
| No LLM | 40.60 | 59.82 | 48.38 | 64.71 | 44.92 | 62.19 | 48.60 | 65.30 | 46.65 | 63.48 | 54.64 | 69.92 | 45.36 | 62.87 | 54.00 | 68.47 |
| **Manhattan distance - ALIGN** | | | | | | | | | | | | | | | | |
| exact | 44.06 | 61.16 | 50.32 | 64.51 | 45.14 | 61.40 | 50.76 | 65.63 | 46.87 | 63.94 | 53.13 | 68.22 | 46.22 | 62.88 | 53.35 | 67.79 |
| what_is | 47.73 | 64.65 | 50.32 | 66.29 | 47.95 | 64.32 | 54.64 | 69.27 | 49.68 | 66.77 | 61.12 | 74.14 | 51.40 | 66.93 | 57.67 | 71.49 |
| describe | 47.08 | 64.31 | 51.40 | 66.87 | 46.87 | 63.98 | 54.43 | 69.11 | 52.48 | 67.89 | 59.83 | 73.42 | 52.27 | 68.00 | 58.75 | 72.02 |
| meaning | 50.54 | 66.93 | 53.78 | 68.79 | 50.54 | 66.38 | 57.02 | 70.92 | 49.89 | 67.42 | **62.42** | **75.67** | 55.51 | 69.99 | 59.61 | 73.21 |
| **Manhattan distance - XLM distilroberta base** | | | | | | | | | | | | | | | | |
| exact | 38.66 | 56.78 | 38.88 | 57.05 | 36.93 | 54.85 | 42.33 | 59.19 | 42.55 | 59.35 | 42.98 | 60.78 | 41.04 | 58.83 | 48.16 | 63.82 |
| what_is | 41.68 | 58.96 | 39.09 | 57.36 | 41.68 | 58.12 | 43.20 | 60.53 | 44.92 | 62.06 | 45.36 | 63.27 | 42.12 | 59.99 | 50.11 | 65.65 |
| describe | 39.96 | 58.52 | 42.98 | 59.69 | 41.04 | 58.48 | 45.57 | 62.19 | 43.41 | 60.34 | 46.44 | 63.53 | 44.06 | 61.57 | 48.60 | 64.99 |
| meaning | 41.47 | 59.13 | 41.04 | 59.03 | 39.96 | 57.64 | 44.28 | 62.26 | 61.79 | 61.76 | 47.73 | 64.63 | 45.57 | 62.30 | **53.13** | **67.83** |

Table 4: Results on phrase-caption retrieval (with and without GPT-3 enhancement) for different captioning models.

| | Similarity | Image source | acc. | MRR |
|---|---|---|---|---|
| **CLIP** | Cosine | Wikidata Images | 34.26 | 50.13 |
| | | Wikipedia Images | 53.26 | 68.14 |
| | Euclidean | Wikidata Images | 33.64 | 49.24 |
| | | Wikipedia Images | 52.17 | 66.95 |
| | Manhattan | Wikidata Images | 33.02 | 48.75 |
| | | Wikipedia Images | 52.82 | 67.25 |
| **ALIGN** | Cosine | Wikidata Images | 31.11 | 47.84 |
| | | Wikipedia Images | 53.26 | **68.44** |
| | Euclidean | Wikidata Images | 30.83 | 47.52 |
| | | Wikipedia Images | **53.48** | 68.40 |
| | Manhattan | Wikidata Images | 31.11 | 47.66 |
| | | Wikipedia Images | 53.26 | 68.27 |

Table 5: Image-to-image retrieval results

provides an explicit direction for potential performance improvements, without such endeavors being computationally prohibiting.

**CoT prompting** results can reveal the internal steps followed by an LLM to reach an answer. In Fig. 2, we present the candidate images for the phrase $t$ "rowing dory", with candidate C serving as the golden image. Captions are extracted using GiT-L captioner with greedy search, thus returning one caption $c_i$ per candidate $i$. We then transform "rowing dory" in a question Q, following the *think* prompt template, as presented in Tab. 2, with captions $c_i, i = A, ..., J$ forming the answer choices.

Consequently, we produce answers via GPT-3.5-turbo with and without CoT, as demonstrated in Tab. 7. In the presented case, CoT encourages

choosing the correct answer C over J, which is preferred by GPT-3.5-turbo when CoT is omitted. Furthermore, CoT elicits the reasoning process of the LLM, thus outputting a detailed explanation regarding the choice of a candidate against semantically similar candidates; for example, C (*a small boat in the water with two oars*) and J (*a row boat with people on it in the water*) are indeed closely related and can be easily confused. As one of the basic VWSD challenges is the fine-grained discrimination between semantically similar candidates, we consider such LLM-derived explanations to be highly informative and accurate for discerning the subtle details between them. More CoT/no_CoT examples are presented in Appendix F.

## 5 Discussion

After the announcement of the Semeval-2023 proceedings, more works achieving state-of-the-art results on VWSD for English have been released. Zhang et al. (2023b) utilize online dictionaries and biencoder architectures for disambiguation and a matching module that selects the most appropriate modality to guide final retrieval. Their approach significantly differs from ours, both in the disambiguation and the ranking part. Patil et al. (2023) rely on an ensemble approach involving fine-tuning CLIP and extracting a joint image-text representation. In their paper they make no usage of knowledge sources but rather resort on more computationally heavy approaches; instead, in our approach we only train a lightweight ranking module, while the usage of computationally demanding modules such as VL retrievers and LLMs is restricted to

| Baseline | LLM-enhance | | Text retrieval features | | | | Image retrieval feat. | | Metrics | |
|:---:|:---|:---:|:---:|:---:|:---:|:---:|:---:|:---:|:---:|:---:|
| $p(i)$ | Prompt | $p(i)$ | Captioner | Embedding | Similarity | Phrase | Embedding | Similarity | Acc. | MRR |
| - | - | - | - | - | - | - | - | - | 63.93 | 76.33 |
| ✓ | - | - | - | - | - | - | - | - | 68.90 | 80.04 |
| ✓ | meaning | - | - | - | - | - | - | - | 73.22 | 82.79 |
| ✓ | meaning | ✓ | - | - | - | - | - | - | 75.16 | 84.13 |
| ✓ | exact | ✓ | - | - | - | - | - | - | 70.41 | 81.10 |
| ✓ | what_is | ✓ | - | - | - | - | - | - | 71.71 | 81.52 |
| ✓ | describe | ✓ | - | - | - | - | - | - | 73.00 | 82.84 |
| ✓ | all prompts | ✓ | - | - | - | - | - | - | 73.87 | 83.96 |
| ✓ | all-except exact | ✓ | - | - | - | - | - | - | 74.30 | 83.80 |
| ✓ | meaning + describe | ✓ | - | - | - | - | - | - | 74.30 | 83.86 |
| ✓ | all-except exact | ✓ | - | - | - | - | ALIGN | manhattan | 76.09 | 85.36 |
| ✓ | all-except exact | ✓ | - | - | - | - | ALIGN | cosine | 76.52 | 85.29 |
| ✓ | all prompts | ✓ | - | - | - | - | ALIGN | cosine | 76.52 | 85.70 |
| ✓ | all prompts | ✓ | BLIP-L-beam | ALIGN | cosine | $t$ | ALIGN | cosine | 77.61 | **85.90** |
| ✓ | all prompts | ✓ | BLIP-L-beam | ALIGN | cosine | all $t_e + t$ | ALIGN | cosine | 77.17 | **86.08** |
| ✓ | all prompts | ✓ | BLIP-L-beam | ALIGN | cosine | $t_{meaning}$ | ALIGN | cosine | 76.52 | 85.63 |
| ✓ | all prompts | ✓ | BLIP-L-greedy | ALIGN | cosine | all $t_e + t$ | ALIGN | cosine | **78.48** | **86.65** |
| ✓ | all prompts | ✓ | GiT-L-greedy | ALIGN | cosine | $t$ | ALIGN | cosine | 77.83 | **86.30** |
| ✓ | all prompts | ✓ | GiT-L-greedy | ALIGN | cosine | $t_{meaning}$ | ALIGN | cosine | 77.39 | **85.92** |
| ✓ | all prompts | ✓ | GiT-L-greedy | ALIGN | cosine | all $t_e + t$ | ALIGN | cosine | 79.35 | 87.23 |
| ✓ | all prompts | ✓ | GiT-L-greedy | ALIGN | cosine | all $t_e + t$ | ALIGN | euclidean | 76.96 | 85.85 |
| ✓ | all prompts | ✓ | GiT-L-greedy | ALIGN | cosine | all $t_e + t$ | ALIGN | manhattan | 76.96 | **86.00** |
| ✓ | all prompts | ✓ | GiT-L-beam | ALIGN | cosine | all $t_e + t$ | ALIGN | cosine | 76.96 | **85.92** |
| LTR of Dadas (2023) (best results) | | | | | | | | | 77.97 | 85.88 |
| SemEval organizers' baseline | | | | | | | | | 60.48 | 73.87 |

Table 6: LTR results using feature combinations as extracted from our previous 4 approaches (baseline, LLM enhancement, text retrieval, image retrieval). ALIGN is employed as the VL retriever. **Colored** instances denote best results overall, while **bold** instances highlight instances that outperform best results of Dadas (2023).

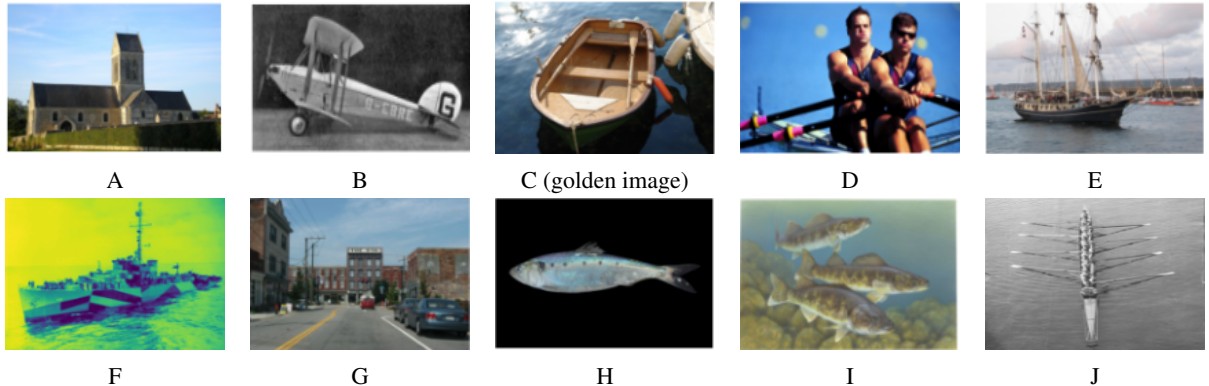

Figure 2: Candidate images for the phrase "rowing dory".

inference. Yang et al. (2023) create a large-scale knowledge base for VWSD based on online dictionaries to acquire the different senses for each ambiguous word. They also train a computationally heavy model in a contrastive setting to tackle VWSD challenges. Overall, our current work de-

| |
|---|
| *Q: What is the most appropriate caption for the rowing dory? Answer Choices: (A) a church with a tall tower and a hedge. (B) an old airplane sitting on top of a runway. (C) a small boat in the water with two oars. (D) the men's rowing team are competing in the men's single sculls. (E) a large black ship in the water near a harbor. (F) a ship is on the water with people on it. (G) a large brick building with a sign on it. (H) a blue fish (sardines) at the australian aquarium. (I) a painting of four fish in the water. (J) a row boat with people on it in the water.* |

| **no_CoT** |
|---|
| Q: What is the most appropriate [...] 
 A: The most appropriate caption for the rowing dory would be (J) a row boat with people on it in the water. **X** |
| **CoT** |
| Q: What is the most appropriate [...] 
 A: Let's think step by step. 
 First, we need to identify the subject of the caption, which is the rowing dory. Then, we need to choose a descriptive and informative caption that accurately represents the image. Out of all the answer choices, (C) "a small boat in the water with two oars" would be the most appropriate caption for the rowing dory as it accurately describes the image and provides relevant information about the boat. Answers (D), (F), and (J) also provide some information about the boat and the people in it, but they either include irrelevant information or are too vague to be an appropriate caption for the image. 
 Therefore, among A through J, the answer is (C) "a small boat in the water with two oars." ✓ |

Table 7: VWSD as QA problem example, with CoT prompting encouraging the selection of the correct candidate. Questions Q are presented truncated.

viates from the aforementioned ones in two main aspects. First, following the inclination of the NLP community towards LLMs we showcase the performance advancements that can be achieved in VWSD via prompting, replacing the usage of online dictionaries for sense disambiguation. With the constant improvement of LLM capabilities we can also expect more advanced phrase enrichment, and thus further boost VWSD performance. Second, we do not train or fine-tune heavy models for the ranking stage: our LTR module requires 20 minutes training on a CPU, thus being a very computationally affordable option for retrieval.

A potential combination of techniques mentioned in Zhang et al. (2023b); Patil et al. (2023); Yang et al. (2023) and ours can yield even more impressive results for VWSD, as long as related computational considerations are relaxed.

## 6 Conclusion

In this work, we provided an extensive foundational experimentation and analysis on the novel Visual Word Sense Disambiguation (VWSD) task. Specifically, we employed several state-of-the-art models for VL retrieval to build strong baselines, showcased the merits of enhancing ambiguous phrases with external knowledge stored in LLMs and achieved competitive ranking results by training a lightweight retrieval module using features extracted from our independent experiments. Moreover, we obtained useful explanations that unveil the reasoning process behind VWSD via Chain-of-Thought prompting. Our results surpassed concurrent implementations and given baselines, while demonstrating valuable insights that can drive future state-of-the-art implementations. We plan to expand our experimentation towards exploring the use of soft prompting to advance knowledge enhancement, and explore the possibility of harnessing large-scale knowledge graphs for further enrichment of ambiguous phrases. Finally, we view explainability aspects of VWSD as a critical direction to be studied as future work.

## Acknowledgments

The research work was supported by the Hellenic Foundation for Research and Innovation (HFRI) under the 3rd Call for HFRI PhD Fellowships (Fellowship Number 5537).

## Limitations

Our current work is accompanied with certain limitations, some of which we plan to address as future work. First of all, due to limited computational resources, we do not resort to very large LMs (>7B parameters) in our machines; however, scaling-up would probably provide advanced knowledge enhancement for short phrases, an assumption that can be strengthened by the advanced results occurring when incorporating the 175B GPT-3 in our experimental pipelines. The usage of GPT-3 and GPT-3.5-turbo was focused on a targeted experimental subset due to high pricing. On the other hand, with respect to this limitation, we chose to distribute our efforts across varying approaches for VWSD rather than focusing to delve into a specific direction (e.g. LLM-based knowledge enhancement) and explore the contribution of larger LMs. Moreover, under this resource limitation, our results can drive researchers that have access to limited computational resources to replicate and extend our analysis, instead of restricting this experimentation to institutions or individual researchers

with higher budgets.

Other than that, the LLM-enhancement technique faces the risks associated with hallucinations and untruthful generation, which cannot be easily detected or resolved, according to related state-of-the-art research (Bang et al., 2023; Mündler et al., 2023; Azaria and Mitchell, 2023; Zheng et al., 2023; Zhang et al., 2023a). Such a shortcoming could negatively impact our results, since we have not investigated whether there are hallucinated or untruthful enhancements, especially since some phrases may require specific domain knowledge to be evaluated (for example, *andromeda tree* is not an everyday term, and even a human needs to consult an encyclopedia to evaluate if a related enhancement is correct). To the best of our knowledge, there has been no open-source tool to accurately and faithfully detect hallucinations, therefore as for now, we cannot resolve this limitation. However, we could possibly mitigate hallucinations and untruthful generation by combining LLM knowledge with knowledge graphs, which can be regarded as more reliable knowledge sources for enhancing VL tasks (Lymperaiou and Stamou, 2023). We will investigate such hybrid techniques in future work.

Finally, we have focused our experimentation on the English language in order to develop a variety of techniques for VWSD rather than testing an applicable subset of them on other languages. Nevertheless, we plan to address this limitation as future work.

## Ethics Statement

Our work involves a newly introduced publicly available dataset released under the CC-BY-NC 4.0 license and can be accessed by any researcher. Throughout this work, we adhered to the fair data usage policy, as required by the dataset creators[4]. We employ language models up to 6.7B parameters which were run on a machine with a 16 GB GPU. Such computational resources are rather easily accessible by most research institutions; therefore, throughout this paper we promote fair and reproducible research, eliminating the need for high-end computational budget. Accessing larger models such as GPT-3 and GPT-3.5-turbo was feasible via their APIs, which do not impose computational limitations from the user's side. The task itself does not involve any obvious risks, as it targets to expand the field of multimodal retrieval. Harnessing language models as knowledge bases imposes the risk of retrieving false or inaccurate information, which however does not induce serious implications in its current usage, given the non-critical nature of this dataset. Overall, we do not anticipate any ethical issues arising from our work.

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

## A Dataset details

In Tab. 8 statistics for VWSD are presented. All train and test samples contain 10 image candidates. The phrase length demonstrates negligible differences, with the vast majority of phrases comprised of 2 words. Data samples and official splits can be found in https://raganato.github.io/vwsd/.

| Split | #Samples | Phrase length | | | |
|-------|----------|---------------|---------|---------|---------|
| | | 1 word | 2 words | 3 words | 4 words |
| Train | 12869 | 0 | 12868 | 0 | 0 |
| Test | 463 | 1 | 445 | 17 | 1 |

Table 8: Dataset statistics

## B Computational resources

In Tab. 9 we analyze the resources used throughout our experiments, as well as the time needed for inference on the entire test set of 463 samples. Regarding captioners, we demonstrate the time needed for one batch of 1000 images. As for LTR, training time refers to the train split exclusively (12869 samples).

## C Additional LLM-enhancement results

**Quantitative results**  In Tab. 12, 13 we present the continuation of Tab. 3 results. LLMs combined with VL retrievers without penalty. Since smaller open-source LLMs do not involve pricing limitations, we attempt to experiment with some additional prompts, mostly paraphrasings of the "describe" prompt, as presented in Tab. 10. To this end, results for OPT-2.7B and BLOOMZ 3B are extended for these new prompts (Tab. 11).

As a general takeaway, we can verify the claim that LLMs in the low-billion parameter scale do not contain the appropriate knowledge to provide high-quality context to semantically enrich ambiguous phrases. For example, according to Tab. 12, OPT-6.7B enhancement leads to scores slightly below the non-enhanced baselines in most cases, irrespectively of the incorporation of penalty in the VL retrieval module. The smaller models of GPT2-XL (1.5B) and BLOOMZ-1.7B showcase some improvements compared to the respective baselines when no penalty is used, but the results remain low compared to the ones reported in Tab. 3. We also observe an interesting variability regarding which prompts induce better enhancement results, with different prompts performing better or worse across models. This verifies that prompts are not transferable, i.e. a prompt that performs well in conjunction with a certain model does not necessarily perform equally well when inserted to another model. In total, all models in Tab. 12, 13 demonstrate comparable performance to each other; even though the reported metrics are not discouraging from the perspective of a potential real use case (e.g.

| Model | Hardware | Time (hours) |
|---|---|---|
| **VL Transformers for retrieval** | | |
| CLIP | GPU - NVIDIA Tesla K40 12GB | 00:10 h |
| ALIGN | GPU - NVIDIA Tesla K40 12GB | 00:08 h |
| CLIP-L | GPU - NVIDIA Tesla K40 12GB | 00:15 h |
| BLIP | GPU - NVIDIA Tesla K40 12GB | 00:20 h |
| BLIP-L | GPU - NVIDIA Tesla K40 12GB | 00:45 h |
| **LLMs for phrase enhancement** | | |
| GPT2 XL 1.5B | GPU - NVIDIA Tesla K40 12GB | 00:30 h |
| OPT 2.7B | GPU - NVIDIA Tesla K40 12GB | 01:45 h |
| BLOOMZ 1.7B | GPU - NVIDIA Tesla K40 12GB | 00:15 h |
| BLOOMZ 3B | GPU 12.8GB | 00:20 h |
| OPT 6.7B | 2 x GPU T4 14.8GB | 02:00 h |
| Galactica 6.7B | 2 x GPU T4 14.8GB | 02:15 h |
| **Image Captioners** | | |
| BLIP (batches with 1000 images each) | GPU - NVIDIA Tesla K40 12GB | ∼02:00 h / 1000 images |
| BLIP-L (batches with 1000 images each) | GPU - NVIDIA Tesla K40 12GB | ∼03:00 h / 1000 images |
| GiT (batches with 1000 images each) | GPU - NVIDIA Tesla K40 12GB | ∼03:00 h / 1000 images |
| GiT-L | GPU - NVIDIA Tesla K40 12GB | ∼04:00 h / 1000 images |
| **Sentence Transformers** | | |
| xlm-r-distilroberta | NVIDIA TITAN Xp 12GB | < 00:08 h |
| stsb-roberta-base | NVIDIA TITAN Xp 12GB | < 00:08 h |
| stsb-distilroberta-base | NVIDIA TITAN Xp 12GB | < 00:04 h |
| stsb-mpnet-base | NVIDIA TITAN Xp 12GB | < 00:07 h |
| all-MiniLM-L6 | NVIDIA TITAN Xp 12GB | < 00:03 h |
| all-MiniLM-L12 | NVIDIA TITAN Xp 12GB | < 00:04 h |
| all-mpnet-base | NVIDIA TITAN Xp 12GB | < 00:06 h |
| multi-QA-distilbert | NVIDIA TITAN Xp 12GB | < 00:06 h |
| multi-QA-MiniLM-L6 | NVIDIA TITAN Xp 12GB | < 00:04 h |
| **LTR** | | |
| LTR training | CPU - 16GB RAM | ∼00:20 h |
| LTR prediction | CPU - 16GB RAM | < 00:01 h |

Table 9: Resources used for our experiments and time needed

| Prompt name | Prompt template |
|---|---|
| would_say | "To describe <phrase> I would say that" |
| could_describe | "I could describe <phrase> as " |
| write_description | "Write a description of <phrase>." |

Table 10: Prompts for phrase enhancement via LLMs.

an image retrieval platform), especially given the difficulty of VWSD, performance achieved using the multi-billion parameter models of GPT3 and GPT-3.5 turbo set higher expectation for related VWSD implementations.

**Qualitative results** We showcase some LLM-enhancement examples on given phrases, accompa-

nied by the label prediction ranking (the leftmost label is the top-1 choice of the VL model). In Fig. 3 candidates corresponding to the phrase "greeting card" are presented, with candidate C being the correct ground truth answer. The baseline predicted label ranking from CLIP is: ['G', 'C', 'D', 'E', 'J', 'B', 'I', 'F', 'H', 'A']; therefore, the golden label is ranked second. We then create the enhancements for "greeting card" presented in Tab. 14, accompanied by their label predictions using CLIP. As concluded by Tab. 14, enhancements can be either beneficial ("describe" prompt enhancement), or not (enhancements apart from "describe") with respect to the prediction. However, by qualitatively evaluating those enhancements, we view them as highly sensible and informative, excluding the "ex-

| | | CLIP | | CLIP-L | | ALIGN | | BLIP$_C$ | | BLIP-L$_C$ | | BLIP$_F$ | | BLIP-L$_F$ | |
|---|---|---|---|---|---|---|---|---|---|---|---|---|---|---|---|
| | | acc | MRR | acc | MRR | acc | MRR | acc | MRR | acc | MRR | acc | MRR | acc | MRR |
| | | | | | | | | With penalty | | | | | | | |
| | **Baseline** | 63.28 | 76.27 | 62.85 | 76.24 | 68.90 | 80.00 | 60.90 | 74.33 | 64.58 | 77.51 | 60.47 | 73.87 | 69.76 | 80.42 |
| OPT | would_say. | 62.53 | 76.07 | 65.17 | 78.02 | 64.15 | 76.92 | 63.07 | 75.22 | 61.99 | 74.91 | 57.02 | 70.74 | 65.44 | 77.33 |
| OPT | could_desc. | 59.83 | 73.70 | 56.99 | 72.31 | **68.68** | **80.41** | 65.23 | 77.44 | 66.09 | 77.44 | 57.45 | 71.88 | 68.25 | 79.57 |
| OPT | write_desc. | 56.44 | 71.82 | 63.37 | 76.52 | 65.44 | 77.68 | 61.99 | 74.65 | 62.42 | 75.29 | 54.64 | 69.49 | 64.79 | 76.70 |
| BLOOMZ | would_say. | 64.79 | 77.49 | 68.03 | 79.21 | 70.19 | 80.66 | 65.23 | 77.45 | 66.31 | 78.16 | 60.69 | 73.92 | 68.68 | 79.81 |
| BLOOMZ | could_desc. | 65.23 | 77.40 | 65.66 | 78.09 | 69.11 | 79.87 | 65.23 | 77.38 | 67.17 | 78.75 | 61.34 | 73.87 | **69.33** | **80.18** |
| BLOOMZ | write_desc. | 65.73 | 77.16 | 66.81 | 79.16 | 68.68 | 80.07 | 62.85 | 76.23 | 64.36 | 77.61 | 58.53 | 72.21 | 67.39 | 79.17 |
| | | | | | | | | Without penalty | | | | | | | |
| | **Baseline** | 59.18 | 72.94 | 60.69 | 74.42 | 65.66 | 77.48 | 57.24 | 72.07 | 61.34 | 75.88 | 57.67 | 71.96 | 65.01 | 77.86 |
| OPT | would_say. | 61.21 | 74.61 | 62.27 | 75.92 | 60.69 | 74.45 | 58.75 | 72.16 | 58.75 | 72.69 | 53.56 | 68.09 | 61.56 | 75.02 |
| OPT | could_desc. | 56.77 | 71.43 | 54.37 | 69.93 | **66.74** | **78.48** | 61.56 | 75.25 | 62.42 | 75.56 | 54.43 | 69.98 | 65.01 | 77.61 |
| OPT | write_desc. | 54.95 | 70.44 | 59.16 | 73.87 | 63.50 | 76.13 | 56.80 | 71.49 | 59.83 | 73.35 | 50.76 | 66.97 | 63.07 | 75.67 |
| BLOOMZ | would_say. | 61.34 | 74.87 | 64.15 | 76.57 | 66.95 | 78.53 | 62.20 | 75.50 | 63.28 | 76.23 | 56.80 | 71.73 | 65.23 | 77.70 |
| BLOOMZ | could_desc. | 61.56 | 74.96 | 61.77 | 75.50 | 66.95 | 78.10 | 62.85 | 75.85 | 65.23 | 77.25 | 57.24 | 71.49 | **67.82** | **79.06** |
| BLOOMZ | write_desc. | 62.26 | 74.85 | 63.56 | 77.02 | 65.87 | 77.75 | 60.91 | 74.43 | 62.42 | 75.68 | 55.51 | 70.35 | 65.01 | 77.40 |

Table 11: Results for zero-shot LLM-based enhancement, as a continuation of results for OPT-2.7B and BLOOMZ-3B (Tab. 3). **Bold** numbers indicate best results for each LLM.

act" phrase enhancement which is truncated (we cannot easily predefine the optimal length for the generated text, and variations in length may result in varying enhancement results). This observation motivates our exploration towards more **explainable solutions**, such as the ones involving Chain-of-Thought prompting.

Another example regarding LLM-enhancement is exhibited in Fig. 4 regarding the ambiguous phrase "suede chamois". The baseline CLIP label prediction is ['A', 'G', 'C', 'E', 'I', 'B', 'F', 'J', 'H', 'D'], ranking the golden candidate G in the second position. Results after GPT-3 enhancement are presented in Tab. 15. We observe that enhancements are highly relevant to the meaning of the ambiguous phrase, and different prompts show satisfactory consistency regarding the enriched phrase they provide. Nevertheless, retrieval results differ for different prompts, when CLIP is used as the VL retriever between the enhanced phrase $t_e$ and the candidate images $i$. This is not an expected behavior in terms of the VL retriever employed, since semantically similar phrases should yield similar (or ideally identical) ranking results. Therefore, we verify the need for retrieval **explanations** and we conclude that the **robustness** of VL models -at least for the task of multimodal retrieval- should be an issue of outmost importance, when such models are designed and deployed.

## D  Additional text retrieval results

In Tab. 16 we present some additional results regarding phrase-caption retrieval using ALIGN as the model to obtain textual representations for phrases and captions. Cosine similarity and Euclidean distance are utilized as a measure of similarity (we also regard distance as a measure of similarity for narrative simplicity). More text-to-text retrieval results employing other VL transformers for text embeddings are presented in Tab. 17 Additionally, results involving purely textual sentence transformers fine-tuned on semantic similarity for phrase and caption embeddings are presented in Tab. 18. In general, the exclusively textual models of Tab. 18 fall behind the performance achieved when VL transformers (Tab. 16, 17), are employed for the textual representations.

## E  Additional LTR results

As a continuation of Tab. 6, LTR results are presented in Tab. 19, which contains results when CLIP is exploited as the VL retriever in place of ALIGN. In both cases, GPT-3 is used as the LLM for phrase enhancement.

As in the case of ALIGN, the incorporation of LLM-enhancement features in the LTR training offers best results. Those results are boosted even further when combining visual features, as occurring

| | | CLIP | | CLIP-L | | ALIGN | | BLIP$_C$ | | BLIP-L$_C$ | | BLIP$_F$ | | BLIP-L$_F$ | |
|---|---|---|---|---|---|---|---|---|---|---|---|---|---|---|---|
| | | acc | MRR | acc | MRR | acc | MRR | acc | MRR | acc | MRR | acc | MRR | acc | MRR |
| **With penalty** | | | | | | | | | | | | | | | | |
| **Baseline** | | 63.28 | 76.27 | 62.85 | 76.24 | 68.90 | 80.00 | 60.90 | 74.33 | 64.58 | 77.51 | 60.47 | 73.87 | 69.76 | 80.42 |
| GPT2-XL-1.5B | exact | 53.88 | 69.51 | 56.32 | 71.12 | 53.35 | 69.57 | 47.52 | 63.67 | 47.95 | 64.49 | 41.90 | 59.79 | 50.54 | 66.96 |
| | what_is | 61.22 | 74.89 | 61.44 | 75.83 | 63.93 | 76.33 | 57.02 | 70.42 | 57.67 | 71.78 | 51.62 | 67.78 | 61.34 | 74.42 |
| | describe | 57.58 | 72.38 | 60.82 | 74.82 | 58.10 | 72.67 | 54.43 | 68.78 | 53.78 | 69.38 | 47.73 | 65.08 | 57.24 | 71.80 |
| | meaning | 60.82 | 75.00 | **65.58** | **77.55** | 64.15 | 76.65 | 58.75 | 72.64 | 59.18 | 73.27 | 52.92 | 68.27 | 63.07 | 75.96 |
| | would_say. | 59.74 | 73.86 | 62.34 | 75.55 | 59.40 | 73.46 | 52.92 | 67.80 | 53.35 | 68.66 | 48.16 | 64.47 | 55.94 | 70.67 |
| | could_desc. | 57.08 | 71.87 | 59.26 | 73.23 | 58.10 | 72.49 | 52.05 | 68.22 | 55.08 | 70.18 | 46.87 | 64.44 | 57.02 | 72.03 |
| | write_desc. | 58.90 | 72.87 | 62.42 | 75.63 | 60.26 | 74.37 | 50.76 | 67.19 | 54.21 | 69.57 | 48.60 | 65.12 | 60.04 | 73.43 |
| BLOOMZ-1.7B | exact | 61.44 | 74.50 | 64.92 | 77.42 | 65.87 | 77.60 | 64.58 | 76.28 | 65.66 | 77.13 | 59.18 | 72.70 | 67.39 | 78.67 |
| | what_is | 63.71 | 76.41 | 66.74 | 78.92 | 65.44 | 77.65 | 63.28 | 75.95 | 65.23 | 77.68 | 58.32 | 72.06 | 66.52 | 78.30 |
| | describe | 64.72 | 77.21 | 64.07 | 77.47 | 68.90 | 80.28 | 61.34 | 75.61 | 64.36 | 77.45 | 58.53 | 72.41 | 66.74 | 78.81 |
| | meaning | 62.63 | 76.38 | 65.01 | 78.17 | 66.74 | 78.27 | 63.50 | 76.44 | 65.44 | 78.29 | 58.53 | 72.50 | 68.25 | 79.74 |
| | would_say. | 63.20 | 76.36 | 66.45 | 78.20 | **70.41** | **80.70** | 64.15 | 76.77 | 66.95 | 78.66 | 57.88 | 72.20 | 68.25 | 79.81 |
| | could_desc. | 64.86 | 77.13 | 63.99 | 77.10 | 69.33 | 79.96 | 62.42 | 75.65 | 65.87 | 78.17 | 58.10 | 72.47 | 68.47 | 79.89 |
| | write_desc. | 62.61 | 76.00 | 65.93 | 78.07 | 68.68 | 79.70 | 60.04 | 73.52 | 64.58 | 77.13 | 57.02 | 71.19 | 67.17 | 78.66 |
| OPT-6.7B | exact | 62.63 | 75.84 | 62.20 | 75.54 | 67.82 | 79.24 | 60.91 | 74.23 | 64.79 | 77.58 | 59.83 | 73.40 | **69.11** | **79.94** |
| | what_is | 61.79 | 75.70 | 64.63 | 77.68 | 64.79 | 77.23 | 61.77 | 75.01 | 63.07 | 76.16 | 57.88 | 71.79 | 65.87 | 77.77 |
| | describe | 64.43 | 76.91 | 65.73 | 78.24 | 65.23 | 77.89 | 61.12 | 74.67 | 63.93 | 77.07 | 56.16 | 71.30 | 66.09 | 78.38 |
| | meaning | 62.17 | 75.84 | 63.61 | 77.19 | 66.74 | 78.47 | 63.28 | 75.93 | 65.44 | 77.43 | 59.83 | 72.96 | 68.03 | 78.75 |
| | would_say. | 63.98 | 76.77 | 61.14 | 75.49 | 68.03 | 78.95 | 61.77 | 74.73 | 60.26 | 74.38 | 55.51 | 70.23 | 65.87 | 77.70 |
| | could_desc. | 58.59 | 73.21 | 61.45 | 74.60 | 65.44 | 77.76 | 63.07 | 75.34 | 64.36 | 76.09 | 58.32 | 71.86 | 65.23 | 77.28 |
| | write_desc. | 60.38 | 74.46 | 60.61 | 75.19 | 64.58 | 77.04 | 57.02 | 71.13 | 61.34 | 74.24 | 53.13 | 68.91 | 64.58 | 76.70 |
| **Without penalty** | | | | | | | | | | | | | | | | |
| **Baseline** | | 59.18 | 72.94 | 60.69 | 74.42 | 65.66 | 77.48 | 57.24 | 72.07 | 61.34 | 75.88 | 57.67 | 71.96 | 65.01 | 77.86 |
| GPT2-XL 1.5B | exact | 49.45 | 66.24 | 53.66 | 69.09 | 51.19 | 67.22 | 44.28 | 61.19 | 45.57 | 62.60 | 35.85 | 55.50 | 47.52 | 64.43 |
| | what_is | 58.61 | 72.59 | 58.61 | 73.62 | 60.91 | 74.25 | 54.21 | 68.22 | 53.56 | 69.03 | 46.00 | 64.08 | 55.94 | 70.71 |
| | describe | 54.76 | 70.24 | 56.49 | 72.14 | 55.94 | 70.55 | 50.97 | 66.20 | 50.11 | 66.62 | 44.71 | 62.21 | 55.51 | 70.07 |
| | meaning | 58.44 | 72.97 | **62.55** | **75.60** | 61.56 | 74.49 | 55.08 | 70.24 | 54.64 | 70.76 | 50.32 | 66.74 | 57.02 | 72.54 |
| | would_say. | 54.51 | 69.99 | 59.34 | 73.54 | 57.45 | 71.77 | 47.52 | 64.76 | 48.16 | 65.10 | 45.36 | 62.79 | 56.16 | 70.50 |
| | could_desc. | 51.85 | 68.28 | 55.77 | 70.84 | 54.64 | 69.73 | 48.81 | 65.20 | 51.19 | 66.97 | 42.76 | 61.44 | 53.13 | 68.74 |
| | write_desc. | 54.51 | 69.99 | 59.34 | 73.54 | 57.45 | 71.77 | 47.52 | 64.76 | 48.16 | 65.10 | 45.36 | 62.79 | 56.16 | 70.50 |
| BLOOMZ 1.7 | exact | 58.82 | 72.23 | 61.66 | 75.05 | 63.28 | 75.56 | 59.18 | 72.96 | 62.85 | 74.99 | 56.37 | 70.60 | 63.50 | 76.20 |
| | what_is | 62.42 | 75.30 | 65.01 | 77.33 | 63.07 | 75.58 | 59.40 | 73.50 | 62.85 | 76.00 | 56.16 | 70.54 | 65.23 | 77.26 |
| | describe | 60.82 | 74.68 | 62.99 | 76.05 | 66.52 | 78.51 | 59.83 | 74.59 | 62.63 | 76.47 | 53.56 | 69.84 | 63.71 | 77.06 |
| | meaning | 58.75 | 73.78 | 64.15 | 77.03 | 64.36 | 76.44 | 60.48 | 74.34 | 61.99 | 76.13 | 56.16 | 70.61 | 65.01 | 77.89 |
| | would_say. | 59.96 | 73.99 | 63.64 | 76.37 | **68.03** | **78.94** | 61.56 | 74.87 | 64.79 | 77.09 | 55.94 | 70.84 | 66.31 | 78.34 |
| | could_desc. | 60.74 | 74.63 | 60.52 | 74.64 | 67.17 | 78.20 | 60.69 | 74.14 | 63.50 | 76.46 | 55.72 | 70.94 | 66.31 | 78.54 |
| | write_desc. | 59.29 | 73.49 | 64.82 | 77.08 | 66.31 | 77.86 | 57.67 | 71.61 | 61.34 | 74.76 | 52.48 | 68.47 | 64.36 | 76.72 |
| OPT 6.7B | exact | 58.75 | 72.63 | 59.61 | 73.86 | 64.15 | 76.57 | 57.24 | 71.96 | 61.12 | 75.83 | 56.80 | 71.40 | 64.79 | **77.66** |
| | what_is | 60.48 | 74.10 | 62.45 | 75.89 | 61.77 | 75.18 | 57.88 | 72.27 | 61.77 | 74.89 | 52.92 | 68.83 | 61.99 | 75.23 |
| | describe | 60.74 | 74.28 | 63.12 | 76.19 | 63.28 | 76.26 | 59.40 | 73.03 | 58.96 | 73.86 | 52.92 | 69.18 | 62.63 | 76.13 |
| | meaning | 59.28 | 73.77 | 62.17 | 76.04 | 63.71 | 76.31 | 52.92 | 74.37 | 61.99 | 75.47 | 55.94 | 70.67 | 65.01 | 77.27 |
| | would_say. | 60.90 | 74.51 | 58.29 | 73.30 | **65.87** | 77.33 | 60.26 | 73.65 | 57.24 | 72.50 | 54.21 | 69.13 | 60.69 | 74.78 |
| | could_desc. | 55.29 | 70.73 | 59.25 | 72.95 | 62.20 | 75.18 | 60.04 | 73.27 | 59.83 | 73.50 | 52.70 | 68.52 | 60.26 | 74.46 |
| | write_desc. | 56.60 | 71.61 | 57.31 | 72.84 | 62.85 | 75.52 | 55.08 | 69.45 | 58.32 | 71.93 | 49.68 | 66.35 | 60.48 | 74.51 |

Table 12: Results for zero-shot LLM-based enhancement. **Bold** numbers indicate best results for each model.

| | | CLIP | | CLIP-L | | ALIGN | | BLIP$_C$ | | BLIP-L$_C$ | | BLIP$_F$ | | BLIP-L$_F$ | |
|---|---|---|---|---|---|---|---|---|---|---|---|---|---|---|---|
| | | acc | MRR | acc | MRR | acc | MRR | acc | MRR | acc | MRR | acc | MRR | acc | MRR |
| **With penalty** | | | | | | | | | | | | | | | | |
| **Baseline** | | 63.28 | 76.27 | 62.85 | 76.24 | 68.90 | 80.00 | 60.90 | 74.33 | 64.58 | 77.51 | 60.47 | 73.87 | 69.76 | 80.42 |
| Galactica 6.7B | exact | 49.66 | 66.59 | 56.24 | 71.44 | 52.48 | 68.45 | 43.41 | 60.82 | 50.32 | 66.95 | 41.90 | 59.68 | 54.00 | 70.23 |
| | what_is | 60.13 | 74.40 | 62.78 | 76.11 | 62.63 | 75.50 | 57.88 | 72.62 | 61.12 | 74.59 | 53.56 | 68.73 | 64.79 | 76.81 |
| | describe | 59.61 | 74.04 | 60.48 | 75.13 | 62.20 | 75.80 | 55.51 | 70.74 | 56.59 | 71.25 | 52.92 | 68.13 | 58.96 | 73.49 |
| | meaning | 60.09 | 74.69 | 60.32 | 74.80 | 62.20 | 75.04 | 61.56 | 74.57 | 61.34 | 74.99 | 52.27 | 68.02 | 61.99 | 75.49 |
| | would_say. | 57.78 | 73.32 | 59.75 | 74.52 | 56.16 | 71.45 | 55.29 | 70.42 | 55.72 | 70.92 | 51.19 | 66.89 | 60.26 | 74.05 |
| | could_desc. | 62.45 | 75.98 | **63.97** | **76.62** | 58.10 | 72.89 | 55.51 | 70.00 | 56.59 | 71.38 | 49.24 | 66.15 | 60.48 | 74.29 |
| | write_desc. | 58.54 | 73.10 | 60.75 | 74.93 | 57.88 | 72.64 | 55.72 | 70.27 | 56.16 | 70.83 | 49.24 | 64.97 | 60.91 | 74.25 |
| **Without penalty** | | | | | | | | | | | | | | | | |
| **Baseline** | | 59.18 | 72.94 | 60.69 | 74.42 | 65.66 | 77.48 | 57.24 | 72.07 | 61.34 | 75.88 | 57.67 | 71.96 | 65.01 | 77.86 |
| Galactica 6.7B | exact | 45.35 | 63.57 | 53.97 | 69.58 | 49.89 | 66.30 | 38.66 | 57.14 | 47.08 | 64.40 | 38.01 | 56.83 | 50.76 | 67.94 |
| | what_is | 56.61 | 71.74 | 59.91 | 73.82 | 60.91 | 74.12 | 55.72 | 70.32 | 58.75 | 72.82 | 52.27 | 67.77 | **61.77** | **74.84** |
| | describe | 56.80 | 71.54 | 58.53 | 73.31 | 59.61 | 73.85 | 53.56 | 69.03 | 80.28 | 67.91 | 50.97 | 66.30 | 54.43 | 70.60 |
| | meaning | 56.69 | 72.21 | 56.24 | 72.63 | 58.96 | 72.64 | 55.72 | 70.79 | 56.37 | 71.65 | 50.11 | 66.57 | 58.32 | 72.90 |
| | would_say. | 54.32 | 70.73 | 56.30 | 72.15 | 52.05 | 68.27 | 51.40 | 67.87 | 50.97 | 67.77 | 46.87 | 63.77 | 54.21 | 70.28 |
| | could_desc. | 60.26 | 74.07 | 62.23 | 75.20 | 54.86 | 69.99 | 54.21 | 68.60 | 54.21 | 69.24 | 45.57 | 63.44 | 58.32 | 72.09 |
| | write_desc. | 55.65 | 71.09 | 57.21 | 72.40 | 54.64 | 70.47 | 50.11 | 66.71 | 51.62 | 67.61 | 43.84 | 61.69 | 57.88 | 72.33 |

Table 13: Results for zero-shot LLM-based enhancement for LLMs containing 6.7B parameters. **Bold** numbers indicate best results for each model.

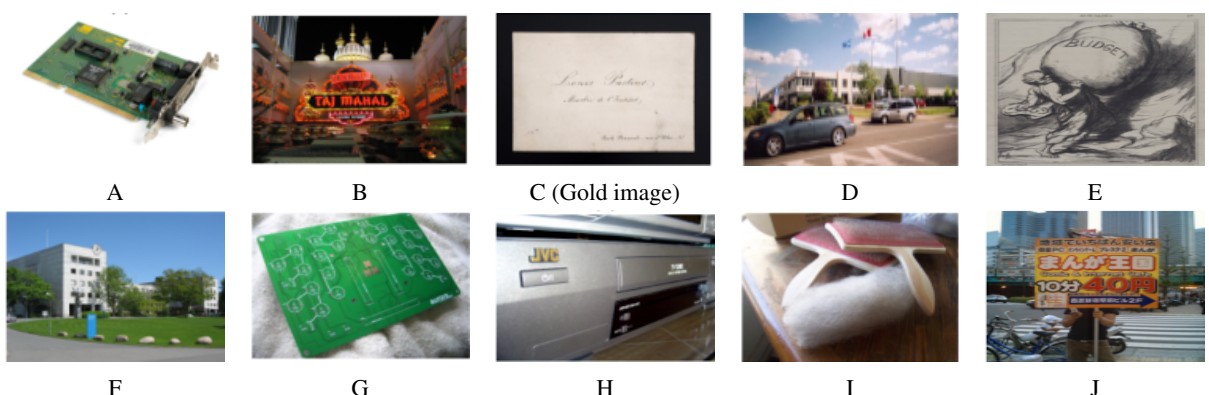

Figure 3: Candidate images for the phrase "greeting card".

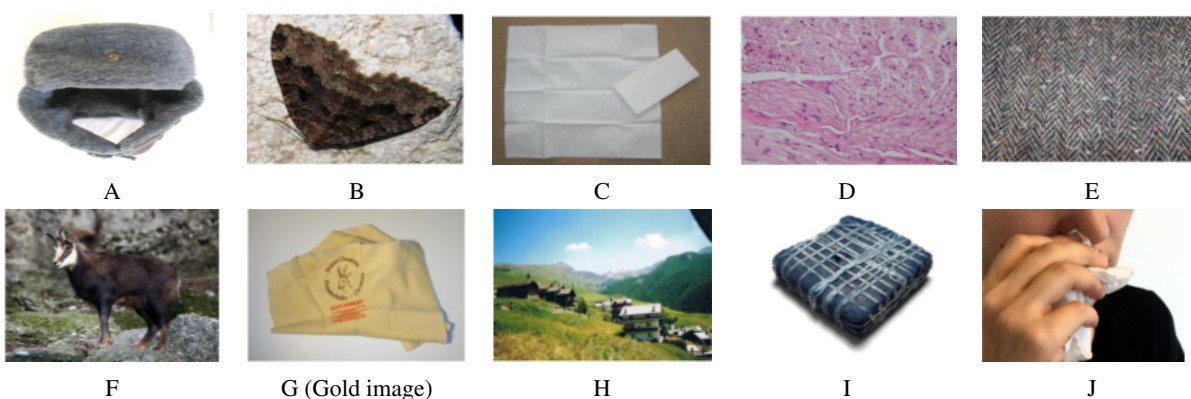

Figure 4: Candidate images for the phrase "suede chamois".

| Prompt name | Enhanced phrase |
|---|---|
| exact | greeting card Dear [Name],I hope this card finds you in good health and spirits. I wanted to take a moment to let you know how much I appreciate you and all that you do. You are an amazing person and I am so lucky to have you in my life.Thank you for always being there for me, for your kind words and support, and for the countless |
| | CLIP predictions: ['G', 'C', 'B', 'E', 'H', 'I', 'J', 'D', 'F', 'A'] |
| what_is | What is greeting card? A greeting card is a card with a message of celebration, congratulations, or other sentiment, typically sent to someone on a special occasion such as a birthday, anniversary, or holiday. Greeting cards often feature artwork, photographs, or illustrations, and are usually printed on high-quality paper stock. |
| | CLIP predictions: ['G', 'D', 'B', 'J', 'E', 'H', 'C', 'F', 'I', 'A'] |
| describe | Describe greeting card. A greeting card is a small, decorated card that is sent to someone to express a sentiment or to celebrate a special occasion. Greeting cards typically feature artwork, a message, and sometimes a small gift such as a token or trinket. They are often sent on holidays, birthdays, anniversaries, and other special occasions. |
| | CLIP predictions: ['C', 'G', 'D', 'B', 'F', 'E', 'I', 'H', 'J', 'A'] |
| meaning | What is the meaning of greeting card? A greeting card is a card sent to someone to express congratulations, best wishes, or other sentiments. It typically includes a message of celebration or goodwill, and may also include an image or design. |
| | CLIP predictions: ['G', 'B', 'D', 'H', 'E', 'J', 'C', 'I', 'F', 'A'] |

Table 14: Enhanced phrases for "greeting card" using GPT-3.

| Prompt name | Enhanced phrase |
|---|---|
| exact | suede chamois Suede chamois is a type of leather that is made from the hide of a sheep. It is known for its soft, velvety texture and is often used to make clothing, shoes, and other accessories. It is also used in upholstery and automotive detailing. |
| | CLIP predictions: ['A', 'G', 'C', 'I', 'E', 'F', 'J', 'B', 'D', 'H'] |
| what_is | What is suede chamois? Suede chamois is a type of leather that has been treated to give it a soft, velvety texture. It is often used for making clothing, shoes, and other accessories. It is also used for cleaning and polishing surfaces, such as cars and furniture. |
| | CLIP predictions: ['C', 'G', 'A', 'I', 'E', 'F', 'J', 'B', 'D', 'H'] |
| describe | Describe suede chamois. Chamois is a type of soft, suede leather that is made from the hide of a chamois, a type of mountain goat. It is known for its softness and absorbency, making it ideal for use as a cleaning cloth or for polishing leather goods. It is also used to make clothing, such as gloves and hats. |
| | CLIP predictions: ['F', 'A', 'G', 'E', 'C', 'I', 'H', 'B', 'D', 'J'] |
| meaning | What is the meaning of suede chamois? Suede chamois is a type of leather that has been buffed to create a soft, velvety texture. It is often used to make clothing, shoes, and other accessories. |
| | CLIP predictions: ['G', 'A', 'C', 'I', 'F', 'E', 'B', 'J', 'H', 'D'] |

Table 15: Enhanced phrases for "suede chamois" using GPT-3.

based on similarity scores between the representations of the 10 candidates $i$ and the representations of the Wikipedia/Wikidata images $i_w$ (both visual representations are extracted using CLIP). In general, the incorporation of more features compared to the VL-baseline only LTR case (first 2 rows of Tab. 19 is helpful, even though the standalone experiments (text-to-text retrieval and image-to-image retrieval) did not produce competitive results (Tab. 4, 5).

# F  QA and Chain-of-Thought examples

Chain of Thought (CoT) prompting provided some interesting insights regarding the explainability of decisions and how it can guide the LLM towards the correct reasoning path. In this section, we provide some additional examples when converting VWSD to a QA problem, comparing prompts with and without CoT. Our presented results highlight particular cases, including the ones where CoT fails; we proceed with an analysis on why this may happen and whether it is an absolutely justified behavior. All presented examples utilize the GiT-L captioner with greedy search. Therefore, one caption per candidate is extracted, and the prompt is constructed appropriately, with each of

| | Greedy | | | | | | | | Beam | | | | | | | |
|---|---|---|---|---|---|---|---|---|---|---|---|---|---|---|---|---|
| | **BLIP** | | **BLIP-L** | | **GiT** | | **GiT-L** | | **BLIP** | | **BLIP-L** | | **GiT** | | **GiT-L** | |
| | acc. | MRR | acc. | MRR | acc. | MRR | acc. | MRR | acc. | MRR | acc. | MRR | acc. | MRR | acc. | MRR |
| **Cosine similarity - ALIGN** | | | | | | | | | | | | | | | | |
| No LLM | 31.97 | 50.41 | 39.52 | 55.00 | 36.93 | 53.51 | 41.04 | 57.67 | 37.37 | 54.29 | 46.65 | 63.21 | 40.39 | 57.04 | 47.30 | 62.59 |
| exact | 34.99 | 51.89 | 38.66 | 54.51 | 38.44 | 54.42 | 43.41 | 58.79 | 39.52 | 56.05 | 52.05 | 66.33 | 42.55 | 58.69 | 46.44 | 61.49 |
| what_is | 38.23 | 55.02 | 41.68 | 56.75 | 41.25 | 57.19 | 46.44 | 61.45 | 40.60 | 57.80 | 56.16 | 70.00 | 47.52 | 62.39 | 51.19 | 65.76 |
| descr. | 39.74 | 56.14 | 42.33 | 57.87 | 43.20 | 58.92 | 48.60 | 63.49 | 42.33 | 59.06 | 57.02 | 70.66 | 50.11 | 64.65 | 52.92 | 67.18 |
| meaning | 36.93 | 53.93 | 40.82 | 56.19 | 41.68 | 57.33 | 44.28 | 60.39 | 41.90 | 58.05 | 54.64 | 69.22 | 48.60 | 63.34 | 50.11 | 65.20 |
| **Euclidean distance - ALIGN** | | | | | | | | | | | | | | | | |
| No LLM | 38.88 | 57.17 | 44.28 | 60.60 | 43.84 | 61.30 | 44.92 | 62.67 | 45.14 | 61.17 | 50.76 | 67.62 | 44.92 | 62.49 | 50.97 | 66.23 |
| exact | 43.63 | 59.57 | 44.06 | 59.99 | 45.14 | 60.86 | 51.19 | 65.38 | 46.22 | 62.40 | 54.86 | 69.37 | 47.08 | 62.81 | 50.32 | 65.78 |
| what_is | 47.95 | 63.83 | 48.60 | 63.73 | 47.52 | 63.73 | 52.92 | 67.55 | 49.46 | 65.40 | 60.69 | 74.05 | 51.84 | 67.11 | 55.51 | 70.31 |
| descr. | 50.32 | 65.58 | 50.11 | 65.21 | 52.05 | 66.61 | 56.59 | 70.03 | 50.11 | 66.46 | 62.20 | 75.35 | 55.51 | 69.68 | 58.75 | 72.23 |
| meaning | 45.57 | 62.18 | 47.52 | 62.91 | 46.65 | 63.80 | 54.21 | 68.32 | 50.54 | 65.78 | 59.40 | 73.46 | 52.48 | 67.97 | 57.67 | 71.28 |

Table 16: Continuation of Tab. 4. Additional results on phrase-caption retrieval (with and without GPT-3 enhancement) for different captioning models using different ALIGN for text embeddings.

the answers for the 10 possible candidates forming answer choices from A to J within the prompt.

**Example 1** In this example, CoT triggers the correct reasoning path towards choosing the correct caption choice (H) "a close up of a metal plate with a pattern of lines." of the corresponding image (Fig. 5), as demonstrated in Tab. 20. The no_CoT prompting case is easily misled by the semantically similar caption (F) that includes the concept "metal" ("a black piece of **metal** with a large black square in the middle"). At the same time, CoT prompting elicit the appropriate behavior of GPT-3.5-turbo, guiding it to distinguish how the phrase "metal steel" differs from the semantically similar one "black piece of metal". This fine-grained capacity is one of the crucial desiderata of VWSD systems, and seems to be somehow connected with CoT reasoning.

**Example 2** There are cases when CoT fails to enhance the given phrase with appropriate context. In the case of Fig. 6, the most relevant image out of the options of Fig. 6 to the phrase "tender embrace" fails to be returned: when GPT-3.5-turbo is prompted with Q, it has a hard time retrieving relevant knowledge, answering that it does not understand where the "tender embrace" phrase refers to, and that more information is needed. By observing the captions comprising the answer choices of the question Q, we confirm that the extracted caption for the golden candidate (C) is sensible and accurate ("a student gets a hug from a student"); thus,

GPT-3.5-turbo cannot connect the phrase "tender embrace" with the correct caption (C), leading to failure, as seen in Tab. 21. On the other hand, when no_CoT prompting is used, GPT-3.5-turbo successfully returns the right answer. We can observe that even though the model does indeed understand the sense of the ambiguous phrase "tender embrace" with respect to the answer choices, it cannot retrieve the appropriate choice under CoT prompting. Therefore, we mark this case as an inherent failure of CoT disambiguation.

**Example 3** Another case that CoT prompting inherently fails, while no_CoT prompting succeeds is presented in Tab. 22 for the phrase "edible sub" and the candidates of Fig. 7. Similarly to the previous case, CoT prompting fails to retrieve the proper caption choice, even though the caption itself is highly relevant and accurate to the candidate image. Its reasoning path outputs some traces of correct reasoning (it mentions the word "sandwich"), even though it cannot proceed further with disambiguation. At the same time, no_CoT prompting succeeds in selecting the right choice (B). It also increases the confidence in its answer by also stating that *"This caption accurately describes the image and highlights the main focus of the picture"*.

**Example 4** There are also cases where both no_CoT and CoT prompts fail to provide the right answer. For example, given the phrase "trotting appendix" and the candidate images of Fig. 8, GPT-3.5-turbo fails to understand the context of the am-

| | Greedy | | | | | | | | Beam | | | | | | | |
|---|---|---|---|---|---|---|---|---|---|---|---|---|---|---|---|---|
| | BLIP | | BLIP-L | | GiT | | GiT-L | | BLIP | | BLIP-L | | GiT | | GiT-L | |
| | acc. | MRR | acc. | MRR | acc. | MRR | acc. | MRR | acc. | MRR | acc. | MRR | acc. | MRR | acc. | MRR |
| **Manhattan distance - CLIP** | | | | | | | | | | | | | | | | |
| No LLM | 36.50 | 56.58 | 41.25 | 58.18 | 36.93 | 55.57 | 39.09 | 56.98 | 41.04 | 58.45 | 44.71 | 61.70 | 41.68 | 59.16 | 43.84 | 60.14 |
| exact | 45.21 | 63.28 | 47.60 | 63.26 | 43.41 | 60.58 | 45.51 | 62.14 | 47.90 | 64.16 | 50.60 | 67.08 | 48.20 | 63.93 | 49.10 | 64.70 |
| what_is | 44.96 | 63.88 | 46.60 | 63.48 | 48.01 | 63.91 | 48.24 | 64.68 | 49.18 | 65.23 | 55.04 | 70.18 | 51.05 | 66.74 | 49.18 | 65.81 |
| describe | 47.87 | 65.46 | 48.82 | 64.84 | 48.34 | 64.90 | 53.08 | 67.90 | 46.92 | 64.50 | 57.35 | 71.63 | 54.03 | 68.52 | 54.98 | 69.84 |
| meaning | 46.78 | 64.70 | 46.56 | 63.47 | 44.35 | 62.07 | 48.34 | 64.99 | 46.56 | 64.01 | 52.55 | 68.57 | 49.00 | 65.45 | 50.55 | 66.27 |
| **Cosine similarity - CLIP** | | | | | | | | | | | | | | | | |
| No LLM | 33.26 | 50.76 | 28.73 | 46.64 | 28.94 | 47.13 | 30.02 | 47.96 | 31.97 | 49.96 | 39.09 | 55.97 | 32.40 | 50.72 | 30.89 | 48.48 |
| exact | 37.72 | 55.34 | 38.02 | 54.82 | 36.83 | 53.74 | 35.33 | 53.07 | 42.22 | 58.73 | 44.31 | 61.92 | 40.42 | 56.99 | 40.72 | 55.99 |
| what_is | 37.24 | 55.91 | 37.24 | 54.50 | 38.41 | 55.03 | 38.17 | 55.40 | 41.45 | 58.42 | 46.14 | 62.39 | 41.92 | 57.88 | 38.17 | 55.33 |
| describe | 39.34 | 57.90 | 45.02 | 60.39 | 44.08 | 59.27 | 41.23 | 57.73 | 40.76 | 58.20 | 50.24 | 66.27 | 46.45 | 62.07 | 41.71 | 58.13 |
| meaning | 38.14 | 56.35 | 34.37 | 52.59 | 35.25 | 53.03 | 39.02 | 55.48 | 39.02 | 56.91 | 44.79 | 61.57 | 42.79 | 58.42 | 38.14 | 54.77 |
| **Euclidean distance - CLIP** | | | | | | | | | | | | | | | | |
| No LLM | 35.85 | 53.63 | 32.40 | 50.51 | 32.83 | 51.61 | 33.69 | 51.45 | 36.93 | 54.85 | 43.20 | 60.12 | 37.58 | 55.30 | 38.88 | 55.42 |
| exact | 43.41 | 61.63 | 44.31 | 60.05 | 42.81 | 59.67 | 43.11 | 60.31 | 46.11 | 62.86 | 52.10 | 67.77 | 44.91 | 61.12 | 48.20 | 63.25 |
| what_is | 44.26 | 62.13 | 44.50 | 60.57 | 43.79 | 60.37 | 44.26 | 61.15 | 47.78 | 63.67 | 55.04 | 69.60 | 48.95 | 64.45 | 46.60 | 62.91 |
| describe | 48.34 | 64.95 | 47.39 | 62.68 | 47.87 | 63.17 | 49.76 | 64.50 | 42.65 | 61.87 | 58.29 | 72.28 | 53.55 | 67.52 | 51.66 | 67.28 |
| meaning | 44.57 | 62.38 | 43.02 | 59.86 | 40.80 | 58.94 | 44.57 | 61.05 | 45.90 | 62.72 | 52.11 | 67.36 | 48.56 | 64.52 | 45.90 | 62.39 |
| **Manhattan distance - CLIP$_L$** | | | | | | | | | | | | | | | | |
| No LLM | 38.01 | 57.95 | 41.90 | 60.49 | 39.96 | 57.78 | 45.36 | 62.24 | 42.12 | 60.12 | 47.52 | 65.43 | 44.06 | 61.20 | 45.36 | 62.95 |
| exact | 44.61 | 62.07 | 45.21 | 62.40 | 43.11 | 60.71 | 46.11 | 63.33 | 48.80 | 64.89 | 51.20 | 67.12 | 43.11 | 61.85 | 49.70 | 65.13 |
| what_is | 43.09 | 61.97 | 47.31 | 64.83 | 44.73 | 62.32 | 50.59 | 66.57 | 49.41 | 65.88 | 54.10 | 70.17 | 49.18 | 65.85 | 51.05 | 67.35 |
| describe | 45.97 | 63.01 | 48.34 | 65.36 | 46.45 | 63.40 | 49.29 | 65.05 | 46.45 | 63.57 | 54.50 | 70.82 | 48.82 | 65.37 | 52.61 | 68.64 |
| meaning | 45.01 | 63.42 | 47.45 | 64.70 | 45.68 | 62.73 | 50.33 | 67.02 | 47.89 | 65.37 | 52.55 | 69.29 | 45.90 | 64.44 | 51.22 | 67.35 |
| **Cosine similarity - CLIP$_L$** | | | | | | | | | | | | | | | | |
| No LLM | 32.61 | 51.28 | 30.67 | 48.63 | 31.97 | 49.48 | 31.10 | 49.30 | 35.64 | 52.69 | 34.56 | 52.87 | 37.80 | 55.75 | 35.42 | 52.64 |
| exact | 35.33 | 53.38 | 35.93 | 53.18 | 36.83 | 54.49 | 34.13 | 52.73 | 43.11 | 58.87 | 36.83 | 54.83 | 42.22 | 59.53 | 38.92 | 55.96 |
| what_is | 36.53 | 55.41 | 37.00 | 54.44 | 38.41 | 55.33 | 37.24 | 55.40 | 41.22 | 57.77 | 39.34 | 56.80 | 44.03 | 61.02 | 37.70 | 55.53 |
| describe | 37.91 | 55.86 | 39.81 | 56.24 | 38.86 | 57.39 | 40.76 | 57.42 | 40.28 | 57.13 | 43.60 | 59.62 | 47.39 | 64.94 | 40.76 | 58.52 |
| meaning | 38.14 | 56.00 | 36.36 | 53.56 | 36.81 | 54.32 | 36.14 | 54.69 | 41.24 | 57.85 | 37.25 | 55.44 | 42.79 | 60.02 | 37.25 | 54.78 |
| **Euclidean distance - CLIP$_L$** | | | | | | | | | | | | | | | | |
| No LLM | 35.85 | 54.44 | 32.40 | 52.91 | 32.83 | 54.65 | 33.69 | 55.50 | 36.93 | 57.62 | 43.20 | 62.20 | 37.58 | 59.01 | 38.88 | 59.30 |
| exact | 43.41 | 58.78 | 44.31 | 57.62 | 42.81 | 57.93 | 43.11 | 60.52 | 46.11 | 62.19 | 52.10 | 64.91 | 44.91 | 60.55 | 48.20 | 62.52 |
| what_is | 44.26 | 59.58 | 44.50 | 60.61 | 43.79 | 60.53 | 44.26 | 62.60 | 47.78 | 62.62 | 55.04 | 67.65 | 48.95 | 63.89 | 46.60 | 63.59 |
| describe | 48.34 | 60.67 | 47.39 | 61.51 | 47.87 | 61.35 | 49.76 | 62.69 | 42.65 | 61.20 | 58.29 | 68.81 | 53.55 | 62.72 | 51.66 | 65.66 |
| meaning | 44.57 | 60.64 | 43.02 | 58.67 | 40.80 | 60.39 | 44.57 | 61.72 | 45.90 | 63.40 | 52.11 | 67.59 | 48.56 | 62.89 | 45.90 | 62.13 |

Table 17: Additional results on phrase-caption retrieval (with and without GPT-3 enhancement) for different captioning models using different VL transformers for text embeddings.

biguous word with respect to the caption choices. Different answers are returned in each case; the no_CoT prompt confidently returns an unsuitable answer ((B) instead of (G)), while the CoT prompt retains an uncertain narrative, stating that more context is needed. Even in that case, CoT prompting is more reliable, not only because it does not indicate an inappropriate candidate as the right answer, but also because it better detects the reason why ambiguity cannot be resolved: a "trotting appendix" horse breed contains certain characteristics, which however were not captured by the captioner. Captions of both images (B) and (G) refer to the semantic "horse" without further details about the animal,

| | Greedy | | | | | | | | Beam | | | | | | | |
|---|---|---|---|---|---|---|---|---|---|---|---|---|---|---|---|---|
| | **BLIP** | | **BLIP-L** | | **GiT** | | **GiT-L** | | **BLIP** | | **BLIP-L** | | **GiT** | | **GiT-L** | |
| | acc. | MRR | acc. | MRR | acc. | MRR | acc. | MRR | acc. | MRR | acc. | MRR | acc. | MRR | acc. | MRR |
| **Manhattan distance - distilroberta base** | | | | | | | | | | | | | | | | |
| exact | 39.52 | 56.71 | 39.09 | 57.43 | 38.66 | 56.94 | 41.04 | 58.99 | 39.52 | 57.34 | 46.22 | 62.29 | 37.37 | 56.04 | 44.49 | 60.81 |
| what_is | 40.17 | 57.96 | 41.04 | 59.18 | 41.25 | 59.24 | 42.33 | 60.21 | 39.31 | 57.56 | 47.52 | 64.63 | 40.39 | 58.35 | 45.79 | 62.07 |
| describe | 38.44 | 57.30 | 40.39 | 58.50 | 42.12 | 59.68 | 44.92 | 61.78 | 42.12 | 58.67 | 47.52 | 64.47 | 40.82 | 58.79 | 48.38 | 64.02 |
| meaning | 38.23 | 57.62 | 42.98 | 60.57 | 42.33 | 60.11 | 44.49 | 61.85 | 43.20 | 59.84 | 45.57 | 63.57 | 40.39 | 59.05 | 47.30 | 63.38 |
| **Manhattan distance - stsb roberta base** | | | | | | | | | | | | | | | | |
| exact | 38.88 | 56.84 | 40.39 | 58.95 | 42.33 | 59.11 | 40.39 | 58.95 | 40.17 | 57.47 | 45.36 | 61.89 | 43.20 | 59.87 | 45.36 | 61.95 |
| what_is | 40.60 | 58.51 | 44.06 | 61.87 | 39.74 | 58.31 | 44.06 | 61.87 | 42.98 | 59.92 | 44.71 | 63.28 | 44.06 | 61.24 | 47.73 | 64.56 |
| describe | 42.55 | 60.03 | 40.17 | 57.58 | 42.12 | 59.63 | 45.14 | 62.17 | 42.98 | 60.04 | 47.95 | 65.14 | 45.14 | 62.48 | 47.73 | 64.66 |
| meaning | 45.14 | 61.37 | 40.17 | 58.30 | 41.04 | 59.02 | 46.65 | 63.33 | 43.63 | 61.02 | 49.46 | 66.34 | 43.20 | 61.61 | 49.89 | 66.24 |
| **Manhattan distance - stsb mpnet base** | | | | | | | | | | | | | | | | |
| exact | 36.50 | 56.07 | 41.04 | 58.82 | 41.25 | 59.10 | 43.63 | 60.66 | 40.60 | 58.75 | 46.87 | 64.34 | 43.41 | 61.26 | 46.00 | 62.31 |
| what_is | 40.39 | 58.92 | 42.33 | 60.80 | 42.55 | 60.90 | 45.14 | 62.40 | 43.41 | 61.05 | 49.68 | 66.47 | 45.14 | 62.24 | 49.46 | 65.19 |
| describe | 42.12 | 60.14 | 43.41 | 60.90 | 44.06 | 61.67 | 49.89 | 65.66 | 43.20 | 61.66 | 47.08 | 65.61 | 47.52 | 64.19 | 50.11 | 66.02 |
| meaning | 41.25 | 59.56 | 43.84 | 62.06 | 44.49 | 62.28 | 50.76 | 66.00 | 44.06 | 61.86 | 50.11 | 67.09 | 47.52 | 64.08 | 50.76 | 66.71 |
| **Manhattan distance - all MiniLM-L6** | | | | | | | | | | | | | | | | |
| exact | 42.55 | 59.89 | 45.36 | 62.33 | 41.04 | 59.53 | 45.14 | 61.81 | 42.98 | 60.78 | 49.24 | 65.45 | 43.41 | 61.58 | 49.24 | 64.92 |
| what_is | 44.49 | 61.49 | 44.71 | 62.44 | 45.79 | 62.51 | 46.22 | 63.57 | 42.55 | 61.10 | 48.60 | 66.17 | 47.52 | 64.16 | 50.76 | 66.74 |
| describe | 43.41 | 61.08 | 44.49 | 62.29 | 41.04 | 59.91 | 49.03 | 65.22 | 43.63 | 61.43 | 50.11 | 66.69 | 42.55 | 61.56 | 49.24 | 66.29 |
| meaning | 42.12 | 60.09 | 45.79 | 63.28 | 44.92 | 62.26 | 45.57 | 63.12 | 43.84 | 61.46 | 49.03 | 66.81 | 45.79 | 63.37 | 51.84 | 67.58 |
| **Manhattan distance - all MiniLM-L12** | | | | | | | | | | | | | | | | |
| exact | 39.52 | 57.99 | 46.22 | 62.19 | 40.60 | 59.30 | 42.98 | 60.83 | 41.04 | 60.16 | 48.81 | 65.19 | 42.55 | 60.67 | 48.60 | 64.91 |
| what_is | 40.39 | 59.02 | 43.63 | 61.88 | 41.68 | 60.14 | 46.00 | 63.36 | 42.76 | 61.36 | 48.81 | 66.42 | 44.71 | 61.97 | 49.46 | 66.10 |
| describe | 40.60 | 59.33 | 44.28 | 61.94 | 41.90 | 60.48 | 47.73 | 64.36 | 42.12 | 61.47 | 48.38 | 66.11 | 44.49 | 62.40 | 51.40 | 67.73 |
| meaning | 40.60 | 59.37 | 43.41 | 62.00 | 43.20 | 61.37 | 47.52 | 64.20 | 42.76 | 61.46 | 50.54 | 67.42 | 46.44 | 63.33 | 49.46 | 66.37 |
| **Manhattan distance - all mpnet base** | | | | | | | | | | | | | | | | |
| exact | 42.55 | 60.63 | 44.71 | 62.80 | 42.98 | 61.06 | 46.22 | 63.00 | 42.98 | 61.72 | 50.76 | 66.84 | 47.08 | 63.90 | 50.32 | 66.39 |
| what_is | 42.55 | 61.56 | 46.22 | 64.69 | 45.79 | 63.20 | 49.24 | 66.13 | 43.41 | 62.80 | 54.43 | 70.45 | 46.65 | 64.33 | 50.97 | 67.57 |
| describe | 42.76 | 61.40 | 46.00 | 64.07 | 43.63 | 61.87 | 48.60 | 65.58 | 44.06 | 62.54 | 52.70 | 68.71 | 45.36 | 63.52 | 52.27 | 68.36 |
| meaning | 43.84 | 62.12 | 49.03 | 66.06 | 45.57 | 62.82 | 48.81 | 65.87 | 44.92 | 63.55 | 54.43 | 70.37 | 48.38 | 65.30 | 50.54 | 67.68 |
| **Manhattan distance - multi-QA distilbert** | | | | | | | | | | | | | | | | |
| exact | 40.60 | 59.86 | 43.41 | 60.81 | 42.33 | 60.13 | 42.55 | 60.73 | 42.98 | 61.53 | 49.03 | 65.71 | 42.98 | 61.51 | 49.24 | 65.35 |
| what_is | 41.47 | 60.90 | 44.06 | 62.50 | 47.52 | 63.33 | 46.22 | 64.01 | 44.71 | 63.42 | 50.11 | 67.47 | 47.73 | 64.83 | 49.46 | 66.49 |
| describe | 41.90 | 60.71 | 42.76 | 61.00 | 47.73 | 63.65 | 46.65 | 63.76 | 42.55 | 61.81 | 49.24 | 66.61 | 46.87 | 64.17 | 50.11 | 66.96 |
| meaning | 42.55 | 61.11 | 44.71 | 62.70 | 45.79 | 62.61 | 46.65 | 64.59 | 45.57 | 63.70 | 50.97 | 68.08 | 45.57 | 63.66 | 50.76 | 67.46 |
| **Manhattan distance - multi-QA MiniLM-L6** | | | | | | | | | | | | | | | | |
| exact | 39.09 | 57.37 | 40.60 | 58.07 | 40.39 | 58.64 | 41.25 | 59.06 | 42.55 | 60.68 | 47.52 | 64.10 | 39.96 | 58.96 | 44.06 | 61.66 |
| what_is | 40.82 | 59.40 | 42.76 | 60.14 | 40.82 | 59.49 | 43.84 | 61.65 | 44.28 | 62.08 | 49.24 | 66.57 | 41.90 | 61.13 | 46.87 | 64.42 |
| describe | 38.23 | 57.35 | 42.33 | 59.58 | 39.74 | 58.84 | 45.14 | 62.05 | 41.68 | 60.51 | 48.38 | 65.32 | 40.82 | 59.65 | 47.30 | 64.50 |
| meaning | 40.39 | 59.04 | 42.33 | 60.46 | 41.68 | 60.05 | 45.57 | 62.92 | 42.33 | 61.35 | 48.60 | 65.93 | 42.33 | 61.26 | 49.89 | 66.55 |

Table 18: Results on phrase-caption retrieval with GPT-3 enhancement for different captioning models using SBERT models for text embeddings.

even though by visually observing the two image candidates, there are obvious differences in terms of patterns and colors between the depicted horses. The rest of the concepts mentioned in the captions such as "field", "fence", "person", "building" do not provide any relevant context to the ambiguous word. Therefore, GPT-3.5-turbo correctly reaches a tie condition where (B) and (G) captions equally

| Baseline | LLM-enhance | | Text retrieval features | | | | Image retrieval feat. | | Metrics | |
|---|---|---|---|---|---|---|---|---|---|---|
| $p(i)$ | Prompt | $p(i)$ | Captioner | Embedding | Similarity | Phrase | Embedding | Similarity | Acc. | MRR |
| - | - | - | - | - | - | - | - | - | 63.93 | 76.33 |
| ✓ | - | - | - | - | - | - | - | - | 68.90 | 80.04 |
| ✓ | - | - | - | - | - | - | - | - | 62.85 | 75.88 |
| ✓ | - | - | - | - | - | - | CLIP | cosine | 70.87 | 81.36 |
| ✓ | - | - | - | - | - | - | CLIP | euclidean | 70.22 | 81.09 |
| ✓ | - | - | - | - | - | - | CLIP | manhattan | 69.78 | 80.95 |
| ✓ | - | - | GiT-L-greedy | CLIP | cosine | $t$ | - | - | 62.85 | 76.08 |
| ✓ | - | - | GiT-L-beam | CLIP | cosine | $t$ | - | - | 63.07 | 76.14 |
| ✓ | - | - | GiT-L-beam | CLIP | euclidean | $t$ | - | - | 62.85 | 75.85 |
| ✓ | - | - | GiT-L-beam | CLIP | manhattan | $t$ | - | - | 62.85 | 76.11 |
| ✓ | - | - | blip-L-greedy | CLIP | cosine | $t$ | - | - | 61.77 | 75.48 |
| ✓ | - | - | blip-L-beam | CLIP | cosine | $t$ | - | - | 62.85 | 75.94 |
| ✓ | all | ✓ | - | - | - | - | - | - | 70.37 | 81.65 |
| ✓ | meaning | - | - | - | - | - | - | - | 65.85 | 78.67 |
| ✓ | meaning | ✓ | - | - | - | - | - | - | 66.52 | 79.21 |
| ✓ | exact | ✓ | - | - | | | | | 65.57 | 78.25 |
| ✓ | what_is | ✓ | - | - | - | - | - | - | 67.45 | 79.55 |
| ✓ | describe | ✓ | - | - | - | - | - | - | 70.14 | 80.75 |
| ✓ | all | ✓ | - | - | - | - | CLIP | cosine | 72.05 | 82.81 |
| ✓ | all | ✓ | blip-L-beam | CLIP | cosine | $t$ | CLIP | cosine | 72.05 | 82.61 |
| ✓ | all | ✓ | GiT-L greedy | CLIP | cosine | $t$ | CLIP | cosine | 70.81 | 82.28 |
| ✓ | all | ✓ | GiT-L-greedy | CLIP | cosine | all $t_e+t$ | CLIP | cosine | 73.91 | 83.53 |
| Our best | | | | | | | | | 79.35 | 87.23 |
| LTR of Dadas (2023) (best results) | | | | | | | | | 77.97 | 85.88 |
| SemEval organizers' baseline | | | | | | | | | 60.48 | 73.87 |

Table 19: LTR results using feature combinations as extracted from our previous 4 approaches (baseline, LLM enhancement, text retrieval, image retrieval). CLIP is employed as the VL retriever.

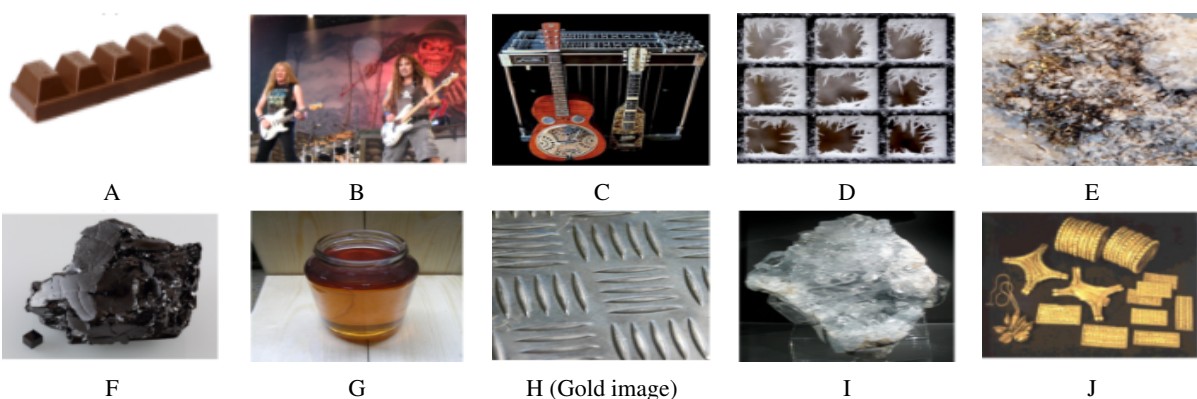

Figure 5: **Example 1.** Candidate images for the phrase "metal steel".

match the ambiguous phrase, therefore failure to provide an answer is totally justified.

**Example 5** Another example that CoT fails due to imperfections in captioning is the one corresponding to the phrase "furry cimarron" and the candidates of Fig. 9. An official description[5] of the term "cimarron" is "wild sheep of mountainous regions of western North America having massive curled horns". However, none of the generated captions contains concepts such as "wild sheep" or "massive curled horns" which are descriptive

[5]https://www.vocabulary.com/dictionary/cimarron

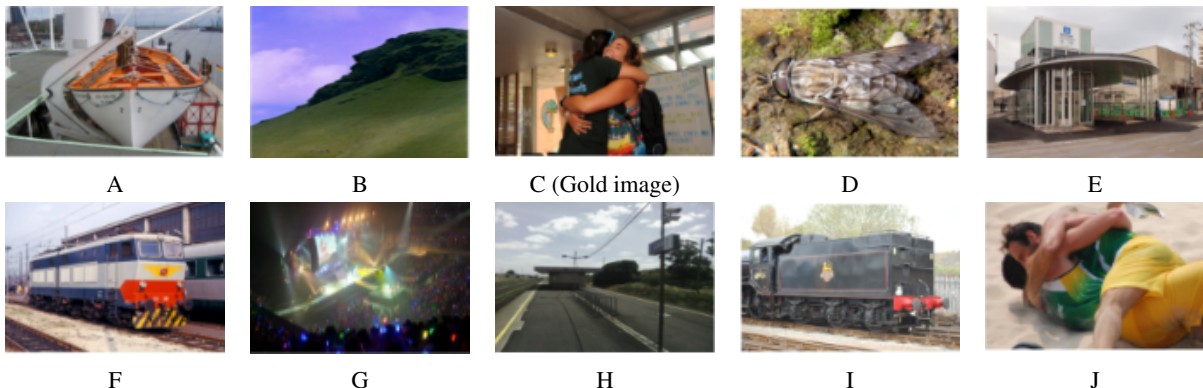

Figure 6: **Example 2.** Candidate images for the phrase "tender embrace".

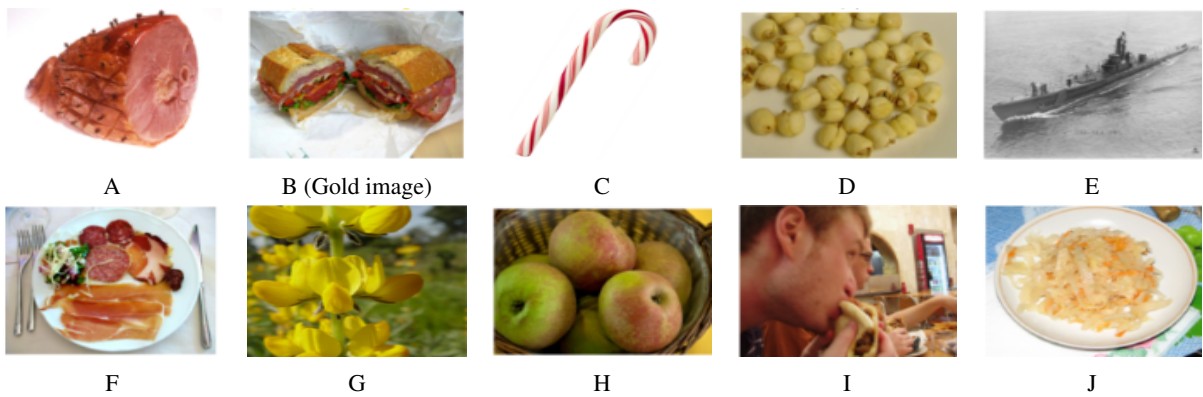

Figure 7: **Example 3.** Candidate images for the phrase "edible sub".

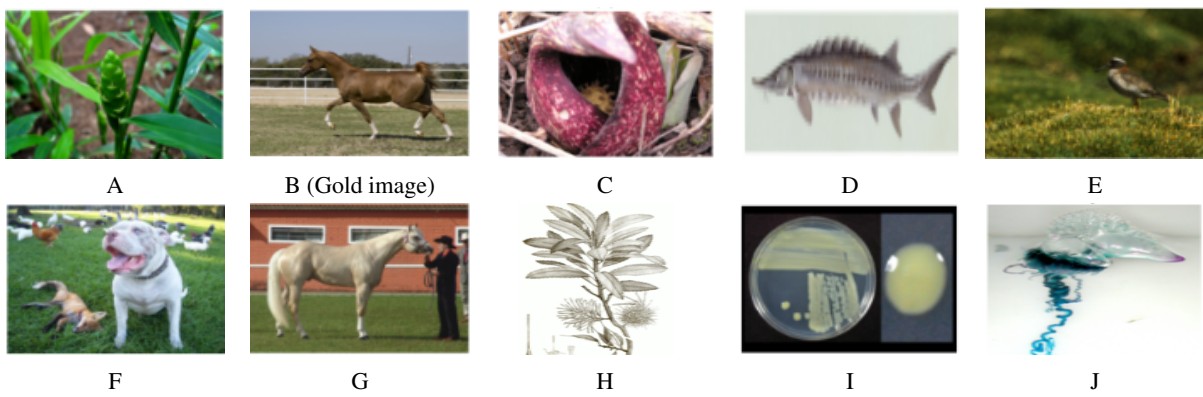

Figure 8: **Example 4.** Candidate images for the phrase "trotting appendix".

---

***Example 1***

*Q: What is the most appropriate caption for the metal steel? Answer Choices: (A) a chocolate bar with three sides (B) [unused0] and [unused0] at the concert in 2007 (C) a guitar and a guitar are displayed in front of a speaker. (D) frosty patterns on a window (E) gold in the rocks - - (F) a black piece of metal with a large black square in the middle. (G) a jar of honey on a wooden table. (H) a close up of a metal plate with a pattern of lines. (I) a large white quartz rock with a clear base. (J) gold jewelry from the late 19th century.*

characteristics of this animal; we assume that the selected captioner does not contain the knowledge needed to fuse the appropriate details in the caption. Therefore, GPT-3.5-turbo correctly avoids to return an answer. Nevertheless, since the concepts "goat" (caption (A)) and "sheep" (dictionary description for cimarron) are semantically related, there should be an inclination towards choosing the answer (A). Erroneously however, GPT-3.5.turbo hints about the answer choice (E) "a rocky cliff face with a body of water in the background", pairing

**no_CoT**

Q: What is the most appropriate [...]
A: The most appropriate caption for the metal steel would be (F) a black piece of metal with a large black square in the middle. **X**

**CoT**

Q: What is the most appropriate [...]
A: Let's think step by step.
First, we need to understand what metal steel is and what its characteristics are. Steel is a hard and strong metal alloy made mainly of iron and carbon. It is often used in construction, machinery, and transportation. Based on this information, the most appropriate caption for metal steel would be (H) a close up of a metal plate with a pattern of lines. This caption describes the texture and appearance of steel, which is often characterized by its distinctive pattern of lines. The other options do not accurately describe steel or its unique qualities.
Therefore, among A through J, the answer is (H) a close up of a metal plate with a pattern of lines." ✓

Table 20: VWSD as QA problem example, with CoT prompting encouraging the selection of the correct candidate. Questions Q are presented truncated.

---

*Example 2*

*Q: What is the most appropriate caption for the tender embrace? Answer Choices: (A) a small boat sitting on top of a dock. (B) a group of people walking on a green hill. (C) a student gets a hug from a student. (D) a large fly laying on a rock in the water. (E) the bus stop at the station (F) a train is parked at a station. (G) a crowd of people watching a concert. (H) a train station with a sign on the side of it. (I) a black and red train on a track. (J) a man laying in the sand on top of a surfboard.*

---

*Example 2*

**no_CoT**

Q: What is the most appropriate [...]
A: The most appropriate caption for the tender embrace would be: (C) a student gets a hug from a student. ✓

**CoT**

Q: What is the most appropriate [...]
A: Let's think step by step.
First, we need to understand what the tender embrace is referring to. Once we know that, we can look at the answer choices and find the one that best matches the subject of the photo. Therefore, without further information about the photo, it is not possible to choose the most appropriate caption. Could you please provide more context or information about the photo?
Therefore, among A through J, the answer is not applicable without more information about the photo." **X**

Table 21: VWSD as QA problem example, with CoT prompting encouraging the selection of the correct candidate. Questions Q are presented truncated.

the terms "mountainous regions" and "rocky cliff face" which are semantically related; the choice (E)

*Example 3*

*Q: What is the most appropriate caption for the edible sub? Answer Choices: (A) a ham ham with a few small spikes on it (B) a sandwich cut in half on a white paper. (C) candy canes - the candy cane company (D) a white plate with a bunch of peanuts. (E) a submarine on the water. (F) a plate of meats, sausages, and salad. (G) yellow flowers on a green field (H) a basket filled with apples sitting on top of a table. (I) a man eating a hot dog in a restaurant. (J) a plate of sauerkraut with carrots and cheese.*

---

*Example 3*

**no_CoT**

Q: What is the most appropriate [...]
A: The most appropriate caption for the edible sub would be (B) "A delicious sandwich cut in half on a white paper." This caption accurately describes the image and highlights the main focus of the picture. ✓

**CoT**

Q: What is the most appropriate [...]
A: Let's think step by step.
First, we need to understand what the "edible sub" is referring to. Is it a sandwich or something else? Once we have this information, we can consider the answer choices and determine which one is the most appropriate caption for the edible sub. Can you provide more information about the "edible sub"?
Therefore, among A through J, the answer is unclear without more information about what the "edible sub" actually is." **X**

Table 22: VWSD as QA problem example, with CoT prompting encouraging the selection of the correct candidate. Questions Q are presented truncated.

*Example 4*

*Q: What is the most appropriate caption for the trotting appendix? Answer Choices: (A) a close up of a plant with a green flower (B) a horse running in a field with a fence in the background. (C) a close up of a flower inside a plant (D) a close up of a fish (E) a bird standing on top of a grass covered field. (F) a dog and a fox laying on the grass. (G) a person holding a horse in front of a brick building. (H) the flowers of the [ unused0 ] (I) a photo of a petri dish with a yellow substance on it. (J) a blue jellyfish with a long tail.*

though does not even describe an animal. Therefore, despite the caption being deficient in properly describing the semantics of the image, the LLM as well seems incapable of inferring the related semantics.

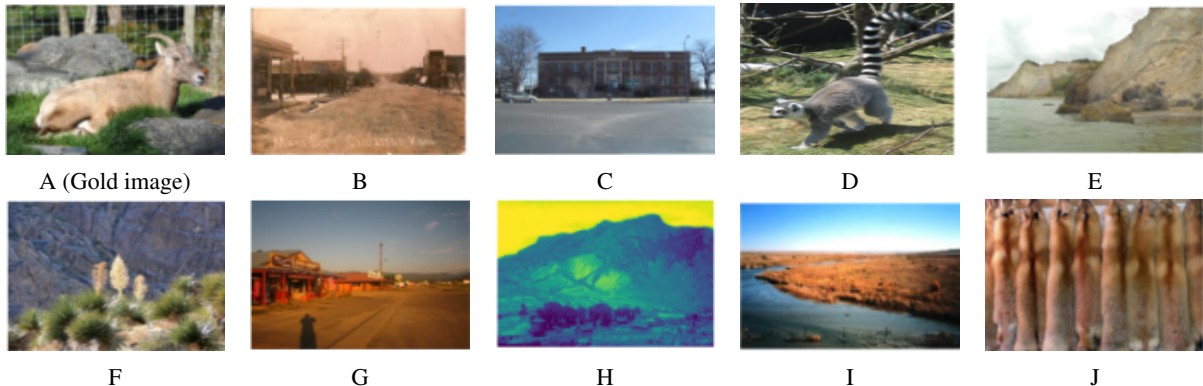

Figure 9: **Example 5.** Candidate images for the phrase "furry cimarron".

| *Example 4* |
| --- |
| **no_CoT** |
| Q: What is the most appropriate [...] A: The most appropriate caption for the trotting appendix would be (B) - "A horse running in a field with a fence in the background." **X** |
| **CoT** |
| Q: What is the most appropriate [...] A: Let's think step by step. First, we need to identify what a trotting appendix is. As an intelligent assistant, I do not have enough information to answer that question. If you could provide more context or clarification, I would be happy to assist you in finding the appropriate caption. Therefore, among A through J, the answer is not applicable." **X** |

Table 23: VWSD as QA problem example, with CoT prompting encouraging the selection of the correct candidate. Questions Q are presented truncated.

| *Example 5* |
| --- |
| Q: *What is the most appropriate caption for the furry cimarron? Answer Choices: (A) a goat laying on a rock in a field. (B) a street that has a lot of buildings on it. (C) a large brick building with a sign on the front. (D) a ring tailed lemur in a zoo (E) a rocky cliff face with a body of water in the background. (F) yucca trees in the desert (G) a street with a person standing in front of it (H) a small town in the mountains. (I) a river running through a field with a fence in the background. (J) red fox furs hang from the ceiling.* |

| *Example 5* |
| --- |
| **no_CoT** |
| Q: What is the most appropriate [...] A: The most appropriate caption for the furry cimarron would be missing from the answer choices. Could you please provide some more information or context about the furry cimarron that could help me suggest a suitable caption?" **X** |
| **CoT** |
| Q: What is the most appropriate [...] A: Let's think step by step. First, we need to understand what the subject of the caption is. The subject is a furry cimarron. Next, we need to think about what would be an appropriate description for the furry cimarron. The options provided do not seem to fit with the subject, except for (E) a rocky cliff face with a body of water in the background. However, this description still does not accurately capture the furry cimarron. Therefore, we need to come up with a new option that specifically describes the furry cimarron. Here are some options: - "A majestic cimarron with a thick coat of fur, basking in the sun" - "A cute and cuddly cimarron taking a nap on a soft bed of grass" - "An independent and free-spirited cimarron exploring the great outdoors" Overall, the most appropriate caption would depend on the context and purpose of the image. Therefore, among A through J, the answer is None of the options are appropriate for describing a furry cimarron. We need to come up with a new option that specifically describes the furry cimarron." **X** |

Table 24: VWSD as QA problem example, with CoT prompting encouraging the selection of the correct candidate. Questions Q are presented truncated.

# G   Model details

We provide the exact Huggingface model cards used in our implementations, or else APIs provided by the creators:

**VL retrieval models**

- CLIP (ViT-base)

  https://huggingface.co/openai/clip-vit-base-patch32

- CLIP-L (ViT-large)

  https://huggingface.co/openai/clip-vit-large-patch14

- ALIGN

  https://huggingface.co/kakaobrain/align-base

- BLIP$_C$ (BLIP with ViT base backbone pre-trained on COCO)

https://huggingface.co/Salesforce/blip-itm-base-coco

- BLIP-L$_C$ (BLIP with ViT large backbone trained on COCO) https://huggingface.co/Salesforce/blip-itm-large-coco

- BLIP$_F$ (BLIP with ViT base backbone pre-trained on Flickr30k) https://huggingface.co/Salesforce/blip-itm-base-flickr

- BLIP-L$_F$ (BLIP with ViT large backbone pre-trained on Flickr30k) https://huggingface.co/Salesforce/blip-itm-base-flickr

**Large Language Models**

- GPT2-XL (1.5B)

  https://huggingface.co/gpt2-xl

- BLOOM-1.7B

  https://huggingface.co/bigscience/bloom-1.7B

- OPT 2.7B

  https://huggingface.co/facebook/opt-2.7b

- BLOOMZ-3B

  https://huggingface.co/bigscience/bloomz-3b

- OPT 6.7B

  https://huggingface.co/facebook/opt-6.7b

- Galactica 6.7B

  https://huggingface.co/facebook/galactica-6.7b

- GPT-3 text-davinci-003 175B (openai API)

- gpt-3.5-turbo (openai API)

**Image captioning**

- BLIP Captions

  https://huggingface.co/Salesforce/blip-image-captioning-base

- BLIP-L Captions

  https://huggingface.co/Salesforce/blip-image-captioning-large

- GiT

  https://huggingface.co/microsoft/git-base

- GiT-L

  https://huggingface.co/microsoft/git-large

**Sentence Transformers** Implementations regarding sentence similarity involve models from Sentence Transformers https://www.sbert.net/.