# OpenReview forum: "Large Language Models and Multimodal Retrieval for Visual Word Sense Disambiguation"
_EMNLP/2023/Conference — EMNLP 2023 Main_

### Official Review · Reviewer_xNwU · 2023-08-02

**Soundness:** 4

**Excitement:**

4: Strong: This paper deepens the understanding of some phenomenon or lowers the barriers to an existing research direction.

**Paper Topic And Main Contributions:**

This paper delves into the topic of visual word sense disambiguation (VWSD) and explores various methodologies to tackle this task. VWSD involves selecting an image from a group of candidate images based on a given phrase containing an ambiguous word. The authors employed different techniques, including:

1. Utilizing LLM models to augment the provided phrase with additional contexts.
2. Investigating unimodal approaches by converting the problem into text-to-text retrieval and image-to-image retrieval problems.
3. Employing a trained learning-to-rank model to combine the strengths of various methods.
4. Applying cot prompting to facilitate the reasoning process in retrieval steps.

The primary findings from this study are as follows:
1. Large-scaled LLMs demonstrate the capability to enhance the context of given phrases effectively.
2. Unimodal approaches do not outperform the performance achieved by the text-enhanced approach.
3. The learning-to-rank model, which integrates enhanced texts and unimodal scores, yields the most favorable results in VWSD.
4. The CoT procedure aids in reasoning over the VWSD task.

**Reasons To Accept:**

This paper focused on the VWSD task introduced in the Semeval-2023 workshop. The authors did comprehensive experiments by utilizing the sota existing pre-trained models through a variant of approaches. Some intuitive approaches, such as text enhancement, text-to-text retrieval, image-to-image retrieval, and training a small ranking model specifically for this task, are explored in this work. The performance of state-of-the-art VLP models, language representation models, and generative models are compared extensively, providing good benchmark results for this task.
Also, the CoT method provides a novel aspect of reasoning over the disambiguation procedure.

**Reasons To Reject:**

Although the experiment of this work is comprehensive, the methodology used in this paper does not provide new insights for this task among the existing works presented in the VWSD workshop:
1. I understand this work is conducted in parallel with the papers presented in the workshop, but the result presented in this paper is not the best for this task.
2. The text-enhancement, text-to-text, and image-to-image approaches are already presented by many teams who joined the workshop.
3. The LTR method used in this task has also been proposed by Dadas et al.
4. The idea of using CoT to explain image retrieval is novel and interesting, but the authors did not use this method to improve the VWSD results. It would be better if the authors can use this information to train a new model, this would be beneficial for this task.
5. The authors reported the results of different models but didn’t discuss the reason why some model achieves better performance than others. The ablation study of the LTR model also did not reveal which component is more important.

**Reproducibility:**

5: Could easily reproduce the results.

**Reviewer Confidence:**

5: Positive that my evaluation is correct. I read the paper very carefully and I am very familiar with related work.

**Typos Grammar Style And Presentation Improvements:**

In Table 3, the pink color that describes the best performance is marked in the wrong cell.

---

> ### Author Rebuttal · Authors · 2023-08-25
>
> We thank reviewer xNwU for their reconstructive comments. We are now presenting our responses to the ‘reasons to reject’:
>
> 1. By the time of submission the paper of Dadas 2023 was the only publically available, and the papers of the competition were only announced one month later. Still, the approach of Dadas 2023 remains on the top-3 after all approaches are considered. In any case, we believe that the LTR module (the best performing in our case, as well as for Dadas 2023) is a promising technique, as it is clear how to boost results (add more features), while being lightweight (fast to train and no requirement for GPU, check Table 9). Other than that, we strongly believe that our paper provides some additional insights other than performance boost: the explainability of VWSD via CoT is addressed for the first time, even if we consider papers published later. Moreover, by investigating different modalities (text-text, image-image, multimodal retrieval) we explore which approach is more viable to drive future directions for VWSD. In any case, we can include results from the rest of the top VWSD papers in the camera ready version.
>
> 2. The multimodal and unimodal retrieval approaches may have been followed by other researchers, but their frameworks were not publically available by the time of EMNLP submission. Therefore, we cannot compare our work with them, and this is why we only compare ourselves with Dadas, 2023. We adopt some ideas presented in their paper, as we mention, but we extend them to achieve better metrics.
>
> 3. Based on the LTR module of Dadas 2023 we do not only achieve an increase in performance (Table 6), but also a significant insight: the incorporation of more features boosts LTR performance, even though some of these features independently (text-text retrieval and image-image retrieval features) are not powerful enough and may even score below the baselines (lines 397-408). Ensembling (meaning that we combine outputs from enhancements derived from different prompts) is even more beneficial, surpassing cases with no ensembling (i.e. only enhancements based on a single prompt are considered). To the best of our knowledge, this analysis was not followed in Dadas, 2023. We believe this analysis can drive future metric boosts for VWSD, for example by crafting additional features or by designing more prompts for enhancement that can be then incorporated in ensembling, therefore it is non-negligible.
>
> 4. CoT was primarily used for explainability, in order to obtain some reasoning insights of how GPT-3.5 reaches an answer, and under which conditions it may fail to do so (Appendix F). We utilized this approach mainly to present another dimension of VWSD apart from performance boosting, as we believe that interpretability issues can be critical in related applications. Of course, extending CoT reasoning and in general VWSD as QA towards metric improvement can be an interesting future direction, beyond the purpose of our current paper.
>
> 5. Regarding the LLM enhancement, we did discuss the reason why larger models present better results, associated with advanced knowledge and/or reasoning capabilities (lines 316-327). As for VL retrieval models, or even image captioners, it is hard or may even not possible to associate performance to a certain aspect, since differences in architecture, pre-training data, training objectives or other factors may impact their final performance. Considering the comparison between multimodal and unimodal approaches, we have concluded that information loss during modality conversion (e.g. captioning to transit to the linguistic modality) and absence of semantically important fine-grained details (e.g. retrieving wikipedia images corresponding to an entity) can be the reason why VL retrieval performs better than text-to-text retrieval (349-352) and image-to-image retrieval (367-370). The ablation of the LTR revealed that the more features are included, the better the final score gets (Tables 6, 19, 20), therefore this is the valuable insights derived from this analysis. Exhausting ablation considering more feature combinations or standalone features (e.g. exclusively captioning features or exclusively image retrieval features) for LTR could have been provided, although we believe it is not as insightful, since it is consistently evident that best results are achieved when more features are incorporated. Overall, in comparison to the LTR approach of Dadas 2023, our paper is the first to provide insights on the ablation of features of the LTR module.

---

### Official Review · Reviewer_xQYd · 2023-08-03

**Soundness:** 4

**Excitement:**

4: Strong: This paper deepens the understanding of some phenomenon or lowers the barriers to an existing research direction.

**Paper Topic And Main Contributions:**

This paper applies a diverse set of approaches to address the visual word sense disambiguation task. The author achieves phrase enhancement by utilizing LLMs on multiple VL retrievers and investigates VWSD as a unimodal problem through conversions to text-to-text and image-to-image retrieval, as well as question-answering (QA) tasks. Furthermore, CoT is developed to guide the generation of explainable answers, and LTR is trained to effectively integrate various modules. Overall, this paper is well-structured with comprehensive experimentation, and the results demonstrate the effectiveness of the proposed method, providing substantial insights for future research.

**Reasons To Accept:**

Although the author did not utilize other widely used image-text datasets for experimentation and focused solely on English data, the primary contribution of this paper lies in providing distinct perspectives for fine-grained image-text retrieval by leveraging limited information in the SemEval dataset. This includes the application of LLMs for phrase enhancement and treating VWSD as a unimodal task through text-to-text and image-to-image approaches. Additionally, the author conducted explanatory explorations of QA reasoning, which may benefit fine-grained text-based retrieval tasks. The experimental results presented in the paper validate the proposed method, and the exposition is coherent and well-organized.

**Reasons To Reject:**

(1) The computation of (d) and (e) seems not be quite meaningful. Mostly, the value of (d) is negative. How to utilize the computations has not been explained in detail.
(2) It is hard to identify a consistent and complete strategy for the task in the Method section. A concrete framework may be needed for this purpose.

**Reproducibility:**

4: Could mostly reproduce the results, but there may be some variation because of sample variance or minor variations in their interpretation of the protocol or method.

**Reviewer Confidence:**

3: Pretty sure, but there's a chance I missed something. Although I have a good feel for this area in general, I did not carefully check the paper's details, e.g., the math, experimental design, or novelty.

---

> ### Author Rebuttal · Authors · 2023-08-25
>
> We thank reviewer xQYd for their reconstructive comments. We are now presenting our responses to the ‘reasons to reject’:
>
> 1. The value of a feature (for example being negative) is not restrictive, as they can act as any other feature fed in a machine learning module trained towards a downstream task (in this case retrieval). For comparative reasons, we designed our features based on Dadas 2023. We assume that this is a fair design since we explicitly compare our LTR module to theirs. The utilization of the calculation of features for the LTR model is described in lines 250-272, as well as in lines 371-388.
>
> 2. Our work focuses on exploring 5 different strategies for performance plus 1 strategy for explainability. Focusing on performance strategies (1-5): strategies 1-4 are independent, therefore consistency is not exactly applicable, since the techniques adopted are not comparable. However, this exploration helps identify to which extent the retrieval modality (visiolinguistic, textual or visual) is important for the final performance. Moreover, strategy 5 incorporates features extracted from strategies 1-4 and achieves best results, therefore constituting the complete framework that a user would adopt for VWSD.

---

### Official Review · Reviewer_tgVD · 2023-08-10

**Soundness:** 4

**Excitement:**

3: Ambivalent: It has merits (e.g., it reports state-of-the-art results, the idea is nice), but there are key weaknesses (e.g., it describes incremental work), and it can significantly benefit from another round of revision. However, I won't object to accepting it if my co-reviewers champion it.

**Missing References:**

N/A

**Paper Topic And Main Contributions:**

This paper introduces multiple baselines for the task of visual word sense disambiguation. The benchmarks are set using state-of-the-art models in LLM or VLPs. These include LLMs for phrase enhancement, transforming images to captions in a sentence-phrase format, converting sentences to images in an image-image format, and employing the 'Learn to Rank' and CoT methods.

While the experimental results appear to be solid and well-founded across different settings, I'm inclined to express reservations regarding the paper's novelty.

The proposed baselines, even with the inclusion of prompts and CoT, come across as rather straightforward and somewhat lacking in complexity. There seems to be a missed opportunity to delve deeper into insightful discussions pertaining to the task's nuances.

Despite variations in the models used (different LLMs and VLPs), they display a strikingly similar pattern. Furthermore, different distance measurements do not seem to significantly alter the outcomes.

**Questions For The Authors:**

1. In Eq1, is the purely similarity-based with a penalty factor still enough to retrieve candidate image?
2. In sec. 6, what is the meaning of A to J, and where do they come from?

**Reasons To Accept:**

- Proposed baselines for tackling the visual word sense disambiguation challenge, employing state-of-the-art models from LLM and VLP categories. Specifically, the authors have experimented with LLMs to enhance phrases, methods to convert images into captions (defined as sentence-phrases), techniques to turn sentences into corresponding images (image-image format), and have also applied the 'Learn to Rank' and CoT approaches.

- The experimental framework appears robust, with results consistently validated across diverse settings.

**Reasons To Reject:**

- The work's novelty seems somewhat restricted, with a noticeable absence of a deep exploration or discussion on the intricacies of the visual word sense disambiguation task itself.
- The proposed baselines, despite their clear delineation, come across as quite direct and predictable. Both the prompt-driven approach and the CoT strategy seem incremental, with an evident lack of an in-depth examination or interpretation of the core task.
- An observed trend is that different LLMs and VLPs exhibit analogous patterns of behavior. Even though there might be variations in the metrics' figures, the general trends and conclusions appear consistent. Distance measurements doesn't seem to significantly affect the outcomes.

**Reproducibility:**

3: Could reproduce the results with some difficulty. The settings of parameters are underspecified or subjectively determined; the training/evaluation data are not widely available.

**Reviewer Confidence:**

3: Pretty sure, but there's a chance I missed something. Although I have a good feel for this area in general, I did not carefully check the paper's details, e.g., the math, experimental design, or novelty.

**Typos Grammar Style And Presentation Improvements:**

Page 2 line 140: B(b)aseline

---

> ### Author Rebuttal · Authors · 2023-08-25
>
> We thank reviewer tgVD for their time and effort they invested in our work. We are going to address the ‘reasons to reject’.
>
> 1. Regarding the lack of ‘deep exploration or discussion on the intricacies of the visual word sense disambiguation task itself’, we denote that this task is only recently introduced (by the time of submission the only publicly available paper was [1]), therefore our work focuses on exploring and introducing a variety of approaches to address task performance, as well as explainability. Nevertheless, in our Introduction section (lines 039-064) we briefly analyze some of the intricacies we recognized throughout our experimentation; especially the explainability issues are not addressed at all in [1]. Of course, further challenges associated with VWSD itself will potentially arise in the future, as more related approaches are introduced. To this end, we view that such challenges deviate from the scope of the current paper, even though we will strongly consider your comment in extensions of our current work.
>
> 2. We introduced a fair baseline exploiting widely used SOTA VL transformers for retrieval, upon which we aim to demonstrate the improvements achieved by various techniques. As we mention (lines 171-174), our prompt engineering strategy for enhancement is based upon prior guidelines designed for different tasks. Thus, we seek to investigate to which extent such intuitive instructions are applicable to VWSD enhancement. The same applies to the usage of CoT prompting. Since our paper is one of the first tackling VWSD we believe it is more fair to step upon prior approaches on other tasks (such as the intuition behind prompt design) and explore how they behave on VWSD before proceeding to more refined techniques. This way, we design a solid and trustworthy way towards exploring and extending VWSD instead of directly jumping to more exotic solutions without proper baselines. An in-depth examination of the core task in conjuction  with the prompt-driven and the CoT approaches can be provided in the camera ready version of the paper, since one extra page is provided.
>
> 3. Regarding the comments ‘ different LLMs and VLPs exhibit analogous patterns of behavior’ and ‘Distance measurements doesn't seem to significantly affect the outcomes’, we view this as a robustness aspect of our work rather than a disadvantage.
> In fact, we consider the presented deviations to be meaningful, e.g. the fact that smaller LLMs present lower MRR/accuracy metrics in comparison to larger ones, denoting that they do not contain the appropriate knowledge or reasoning capabilities to achieve adequate enhancement (lines 316-327). Regarding the metrics, if different metrics resulted in different outcomes, it would require further experimentation to define under which circumstances a metric performs better or worse, and it would be hard to trust a single metric to utilize.
>
> Answers to questions:
>
> 1. Eq 1 corresponds to the VL retrieval Baseline, which appears in the ‘Baseline’ row of Table 3. Therefore, we refer reviewer tgVD to the corresponding row of Tab. 3 (check the ‘with penalty’ part of Tab. 3). Different VL models for retrieval demonstrate slightly different capabilities in retrieving the correct candidate according to the presented MRR/accuracy scores.
>
> 2. A to J is the enumeration of the 10 candidate images, following the enumeration style (capital letters) followed in [2].
>
> [1] Sławomir Dadas. 2023. Opi at semeval 2023 task 1: Image-text embeddings and multimodal information retrieval for visual word sense disambiguation.
>
> [2] Takeshi Kojima, Shixiang Shane Gu, Machel Reid, Yutaka Matsuo, and Yusuke Iwasawa. 2023. Large language models are zero-shot reasoners.

---

### Meta-Review · Area_Chair_UcfJ · 2023-09-14

**Recommendation:** 5

**Metareview:**

This paper looks at the task of visual word sense disambiguation, where the goal is to retrieve an image that matches a given ambiguous word within a particular context. Several approaches are explored, including utilising large language models (LLMs) as an external knowledge source. The reviewers agreed that the work was sound, with comprehensive experiments and comparisons to state-of-the-art baseline systems on this task. The reviewers also commented positively on the use of LLMs for this task. The reviewers raised (minor) concerns about novelty and the question of why some of the models perform better than others. But in general, the work was found to be sound and exciting.

---

### Decision · Program_Chairs · 2023-10-07

**Decision:**

Accept-Main

**Comment:**

This paper looks at the task of visual word sense disambiguation, where the goal is to retrieve an image that matches a given ambiguous word within a particular context. Several approaches are explored, including utilising large language models (LLMs) as an external knowledge source. The reviewers agreed that the work was sound, with comprehensive experiments and comparisons to state-of-the-art baseline systems on this task. The reviewers also commented positively on the use of LLMs for this task. The reviewers raised (minor) concerns about novelty and the question of why some of the models perform better than others. But in general, the work was found to be sound and exciting.